# Immunobiological Properties and Clinical Applications of Interleukin-38 for Immune-Mediated Disorders: A Systematic Review Study

**DOI:** 10.3390/ijms222212552

**Published:** 2021-11-21

**Authors:** Abdolreza Esmaeilzadeh, Nazila Bahmaie, Elham Nouri, Mohammad Javad Hajkazemi, Maryam Zareh Rafie

**Affiliations:** 1Department of Immunology, School of Medicine, Zanjan University of Medical Sciences, Zanjan 4513956111, Iran; 2Cancer Gene Therapy Research Center (CGRC), Zanjan University of Medical Sciences, Zanjan 4513956111, Iran; 3Immunotherapy Research & Technology Group, Zanjan University of Medical Sciences, Zanjan 4513956111, Iran; 4Department of Allergy and Immunology, Faculty of Medicine, Graduate School of Health Science, Near East University (NEU), Nicosia 99138, Cyprus; nazila69bahmaieimmunology@gmail.com; 5Pediatric Ward, Department of Allergy and Immunology, Near East University affiliated Hospital, Nicosia 99138, Cyprus; 6Serology and Immunology Ward, Clinical Diagnosis Laboratory, Private Baskent Hospital, Nicosia 99138, Cyprus; 7Network of Immunity in Infection, Malignancy and Autoimmunity (NIIMA), Universal Scientific Education and Research Network (USERN), Tehran 1419733151, Iran; 8School of Paramedicine, Zanjan University of Medical Sciences, Zanjan 4513956111, Iran; elunr.nri@gmail.com; 9Shahid Beheshti University Affiliated Hospital, Zanjan University of Medical Sciences, Zanjan 4513956111, Iran; 10School of Medicine, Zanjan University of Medical Sciences, Zanjan 4513956111, Iran; Mohammad_hajkazemi1996@yahoo.com (M.J.H.); Maryam.rafie72@gmail.com (M.Z.R.)

**Keywords:** biomarker, clinical applications, diagnosis, immunobiology, immunopathophysiology, interleukin-38

## Abstract

Exponential growth in the usage of “cytokines” (as seroimmunobiomarkers) has facilitated more accurate prognosis, early diagnosis, novel, and efficient immunotherapeutics. Numerous studies have reported immunopathophysiological and immunopathological processes of interleukin-38 (IL-38). Therefore, in this systematic review article, the authors aimed to present an updated comprehensive overview on the immunobiological mechanisms, diagnostic, and immune gene-based therapeutic potentials of IL-38. According to our inclusion and exclusion criteria, a total of 216 articles were collected from several search engines and databases from the January 2012 to July 2021 time interval by using six main keywords. Physiologic or pathologic microenvironments, optimal dosage, and involved receptors affect the functionalities of IL-38. Alterations in serum levels of IL-38 play a major role in the immunopathogenesis of a wide array of immune-mediated disorders. IL-38 shows anti-inflammatory activities by reduction or inhibition of pro-inflammatory cytokines, supporting the therapeutic aspects of IL-38 in inflammatory autoimmune diseases. According to the importance of pre-clinical studies, it seems that manipulation of the immune system by immunomodulatory properties of IL-38 can increase the accuracy of diagnosis, and decipher optimal clinical outcomes. To promote our knowledge, more collaboration is highly recommended among laboratory scientists, internal/infectious diseases specialists, oncologists, immunologists, diseases-specific biomarkers scientists, and basic medical researchers.

## 1. Introduction

Although morbidity and mortality rates of immune-mediated disorders have been efficiently diminished by the use of several cutting-edge technologies (such as novel vaccines), some obstacles in the diagnosis, prognosis, and treatment of immune-mediated disorders have not been tackled yet. The chronicity of these types of disorders has still remained a problematic challenge for the health system. In addition, patients with immune-mediated disorders experience poor lifestyles and suffer from complex clinical manifestations in their life expectancy due to the late or inaccurate diagnosis and poor prognosis. Presumably, these problems have led health system managers to conduct more efficient research to overcome these obstacles [1,2,3,4].

Along with substantial signs of progress in medicine and basic medical sciences, no one can deny the importance of effective immune responses in the immunopathophysiology involved in the microenvironment of immune-mediated diseases. A wide array of evidence suggests that a deviation in the immunological pathways or their subsequent signaling can lead to a malfunction/dysfunction in the immune system, dysregulated production of immunological agents, and severe host tissue damages [1,2,5].

Altogether, these points have led basic medical scientists toward conducting research aimed at comprehensive investigation of cellular and molecular immunology and mechanisms involved in the human immune system. These types of studies help us in order to reach the most effective therapeutics. In this way, the adaptive immune system acts as a potent inflammatory network for the secretion of certain immune-derived factors. The majority of immune-mediated diseases are affected by these seroimmunobiomarker messengers, for example, “cytokines”. Therefore, targeted manipulation of the immune system could pave the road for compensatory immune cascades. As such, the results of in-depth studies on molecular immunobiology of cytokines may provide a clinical basis for further investigations of the therapeutic applications of cytokines in immune-mediated diseases [6,7].

Among these seroimmunobiomarker messengers, members of the Interleukin-1 (IL-1) superfamily are essential players in the interactions of both innate and adaptive immune systems [8,9,10]. In total, the IL-1 superfamily encircles four main subfamilies (IL-1, IL-18, IL-33, and IL-36) with a total of 11 members. All of these members play a key role in the initiation and regulation of the immune responses in immune-mediated diseases. One of the most recently recognized members of this family is IL-38 (IL-1F10), mainly representing anti-inflammatory activities [11,12,13,14,15,16,17,18]. Recently, different studies have demonstrated multiple roles of IL-38 in several immune-mediated conditions (Figure 1). Thus a comprehensive understanding of the immunobiological mechanisms of IL-38 and its immunopathophysiological effects on different diseases could successfully lead to the development of IL-38 gene-based immunotherapeutics in clinics [19,20,21,22].

In this review, we first investigate the immunobiology of IL-38, mainly including the IL-38 receptors, mechanism of signaling, and functional roles of IL-38. Second, we summarize the role of IL-38 in the immunopathophisiology involved in multiple immune-mediated diseases. Finally, we discuss the proficiency of IL-38-based therapeutic strategies such as IL-38 cytokine therapy and IL-38 immune gene therapy as new promising seroimmunobiomarkers for the treatment of several diseases. According to the results acquired from this study, there are more opportunities for developing optimistic insights into more collaborations between physicians, clinical specialists, and laboratory scientists. In addition, these studies encourage physicians and clinical specialists to accredit their clinical decisions with laboratory-based data for improved clinical outcomes. As such, the results of this study could also highlight the importance of pre-clinical and basic medical studies to reach optimal clinical outcomes for clinicians. In addition, clinical administration of seroimmunobiomarkers like cytokine-based therapies can hopefully increase life expectancy for patients due to the reduced side effects when compared to other conventional approaches.

## 2. Methodology

This present systematic literature review study was performed according to the Preferred Reporting Items for Systematic reviews and Meta-Analyses (PRISMA) statement guidelines (https://www.equator-network.org/reporting-guidelines/prisma/, accessed on 13 August 2021) (Figure 2).

### 2.1. Literature Search Strategy, and Screening Process

An electronic comprehensive literature search was conducted with the time interval starting from January 2012 to July 2021 by using six main keywords (Biomarker, AND Clinical Applications, AND Diagnosis, AND Immunobiology, AND Immunopathophysiology, AND Interleukin-38), and five complementary ones (Immunomodulation, AND Inflammatory Diseases, AND Interleukin-1 Superfamily, AND Interleukin-36 sub-family, AND IL-38). The selection was based on our inclusion and exclusion criteria, which are mentioned in the next subsection.

In order to find potentially eligible resources, the screening process for our search strategy was independently conducted by N.B. in three main and one backward steps (on the references and bibliographies of the included articles) according to all of the considered keywords and inclusion/exclusion criteria. Any uncommon points or disagreements were referred to the corresponding author for consultations (Figure 2).

### 2.2. Inclusion and Exclusion Criteria

According to the aim of this study, all of the published original (experimental and non-experimental) and review (mini-reviews, best evidence, narrative reviews, systematic reviews, systematic reviews, and meta-analysis), comparative, cross-sectional, cohort, observational, commentary, letters to the editors, editorial, and case reports/case studies were considered in the formats of full-text/abstract, section of book, chapter, and conference papers. Studies only in English language (or abstracts published in English language) were included. Studies that involved human and animal subjects were all considered. All of the studies aimed at the investigation of IL-38 in whole blood, and other types of samples such as serum, plasma, cerebrospinal fluid, and saliva samples were included. Comparative experimental studies that investigated the expression of IL-38 in control and patient samples were also included as were studies with no control group. In order to report the immunomodulatory effects of IL-38 without any bias due to the usage of adjuvants, vaccines, other stimulators or inhibitors, studies that investigated the secretion of IL-38 after usage of these agents were excluded. Studies with a lack of data related to the structural properties and immunomodulatory characteristics were not analyzed. Studies with no prognostic/diagnostic/therapeutic values and with no serological/molecular measurement of IL-38 were excluded. Research with insufficient/undefined or ambiguous data about IL-38 biology, applications, or expression were also excluded.

### 2.3. Data Extraction

Four authors (N.B., E.N., M.J.H., and M.Z.R.) independently performed the data extraction process and made forms to collect study characteristics (author name, publication date, type of study, study design, and investigated diseases). According to all of the aforementioned inclusion and exclusion criteria, studies aimed at the exploration of the immunobiological features and clinical applications of IL-38 expression were included. In the case of overlapping data or several published reports from the same study, the authors tried their best to cover the details and present the most complete and necessary data.

### 2.4. Quality Assessment and Bio-Statistical Analysis

According to the structure and type of this study (systematic literature review), no test or scale and no bio-statistical approaches were carried out.

### 2.5. Ethical Statement

According to the structure and type of study (systematic literature review), there was no need to register this project with the Research Ethical Committee (REC). It is worth mentioning that all of the data that support the findings of this study are openly available in the context of this manuscript.

## 3. Results

### 3.1. Immunobiology of IL-38

IL-1 superfamily members including IL-38 are involved in immune regulation, host defense, and inflammatory cascades. According to immunohistochemistry (IHC) analysis, IL-38 is secreted from many cells including keratinocytes, monocytes, and macrophages, and is found in many tissues such as tonsil, skin, spleen, thymus, heart, fetal liver, and placenta. Recently, it has been demonstrated that IL-38 is mainly found in the cytoplasm of human keratinocytes [23,24,25,26,27,28]. Secretion of IL-38 from apoptotic cells is not properly recognized. It has only been demonstrated that an impaired inflammation is considered as the sequel of IL-38 secretion from apoptotic cells. In other words, secretion and processing of IL-38 to truncated forms of IL-38 can lead to an immunoregulation by apoptotic cell-dependent signaling, eventuating to the resolution of inflammation or autoimmunities [12,13,29,30]. The IL-1 superfamily consists of several cytokines that are categorized into different groups based on their effects on receptors. In this sense, some of these cytokines are generally known as agonists (e.g., IL-1α, IL-1β, IL-18, IL-33, IL-36α, IL-36β, and IL-36γ), whereas others are as considered as receptor antagonists (e.g., IL-1RA, IL-36RA, and IL-38). Another member, IL-37, is an anti-inflammatory agent (Figure 3) [31,32,33,34,35,36,37,38,39,40,41,42,43,44,45,46,47,48,49,50,51,52].

There is another classification for the IL-1 superfamily, which is based on pro-inflammatory and anti-inflammatory activities of members, dividing the IL-1 superfamily into two distinct groups. Among the members of the IL-1 superfamily, IL-38, IL-1Ra, IL-36Ra, and IL-37 are in the anti-inflammatory group [53]. IL-38 is the most recently known member of the IL-1 superfamily, originally named IL-1F10, IL-1HY2, FKSG75, IL-1-theta, MGC11983, MGC119832, and MGC119833 in various nomenclature systems, in which IL-1F10 is the most practical one. IL-33 and IL-36 subfamilies as well as IL-37 and IL-38 are discovered by an in silico gene identification system [54,55,56,57].

The genomic position of IL-38 in a cluster on chromosome 2 is between IL-36N and IL-1RN encoding genes (ch2q13_14.1). IL-38 shows 43% and 41% homology with IL-36N and IL-1RN, respectively. Biologically, IL-38 indicates a lower affinity in comparison with IL-1RA and IL-1β [26,27,53,58,59,60,61,62]. The IL-1F10 gene has five exon regions, encoding the 152 amino acid protein with 17–18 kDa molecular weight [54]. IL-38 is found in all types of vertebrates with developed immune responses [37]. Regarding purification and expression of IL-38 in bacterial systems, there is one study conducted by Yuan et al. who successfully achieved this. They used a prokaryotic expression vector (named pET-44) to evaluate the expression of IL-38 in a one-step Ni^2+^-agarose affinity chromatography in a relatively large quantity of a C-terminus tagged IL-38 [27].

As previously mentioned, IL-38 is expressed in the skin, spleen, tonsils, thymus, heart, and liver tissues of the fetus in healthy people [63,64]. However, lower levels of this cytokine in these organs do not induce a specific role in the immunity processes [63]. It has been demonstrated that major functions of IL-38 are apoptosis-mediated phagocyte regulation [65], and inhibition of the induction of responses related to T Helper17 (TH17) cytokines. Additionally, IL-38 functions as an antagonist for IL-36, explaining the anti-inflammatory effects of IL-38 on the immune cells [53,66,67]. As IL-38 shows homology with IL-1Ra (an IL-1R1A) and with IL-36RA, it can be found that IL-38 has antagonistic functions with IL-1R1 and IL-36, respectively [20]. Eventually, pre-inflammatory and anti-inflammatory effects of IL-38 were considered as dose-dependent immunological responses. As a prime instance, results acquired from a study on an IL-38 knock-out murine model of asthma (KO) showed that severe clinical manifestations are associated with upregulation in the expression of IL-1β and IL-6 genes in the joints compared to the control ones, meanwhile, recombinant IL-38 could not be able to inhibit arthritis progression [68,69].

#### 3.1.1. Activation of the IL-1 Superfamily Members

Some members in the IL-1 superfamily are activated and matured by a variety of proteases including caspase 1, elastase, cathepsin G, and caspase 8 [54]. Biologically, IL-38 is released from cells without peptide signaling. An inflammasome caspase-1-independent pathway is triggered for the activation of stimuli-induced intracellular IL-1 superfamily cytokines including IL-1α, IL-1Ra, IL-33, IL-36α, IL-36β, IL-36γ, IL-36RA, IL-37, and IL-38 [54,61]. From functional aspects, although the full-length molecule of IL-38 is bioactive, it lacks an integrated N-terminus to function as much as a processed cytokine [60], furthermore, there is even no nuclear localization signal (NLS) on IL-38 for caspase 1 and proteases [9,23].

#### 3.1.2. IL-38 Receptors

Receptors of IL-1 superfamily members vary from IL-1R1 to IL-1R10. Following the interaction with specific ligands, signaling processes result in the activation of several immunological pathways including nuclear factor-kappa B (NF-κB) or JNK/activator protein-1 (AP-1) [54,70]. Three are several receptors interacting with IL-38. Among them, there are three receptors that have shown great biological potentials to interact with IL-38 including IL-36R, IL-1R1, and interleukin-1 receptor accessory protein-like 1 (IL-1RAPL1) [54].

It is worth-mentioning that there were ambiguities in the immunobiological functions of IL-38 when it was first discovered. On one hand, by binding to IL-36R and preventing the interleukin-1 receptor accessory protein (IL-1RAcP) from functioning, IL-38 acts as an antagonist for pro-inflammatory properties of IL-36m especially in peripheral blood mononuclear cells (PBMCs) (Figure 4). On the other hand, the dual functions of IL-38 introduce IL-38 as an agonist for inflammatory responses, especially in dendritic cells (DCs). To be more precise, due to the existence of a wide array of documents related to the inhibitory properties of IL-38 on IL-36 [71], IL-38 is mainly considered to be a potent anti-inflammatory cytokine. This anti-inflammatory function of IL-38 is deeply rooted in the suppression of NF-κB and mitogen-activated protein kinases (MAPKs) signaling by MyD88 (Figure 4), which is normally found in some fungal infections (Candidiasis), psoriasis, and rheumatic diseases. It is noteworthy that IL-38 acts as a natural inhibitor of candida-induced production of IL-2 and IL-17 [23,26,44,52,60,67,72,73,74]. Additionally, it has been demonstrated that IL-38 can play an indisputable role in the inhibition of β-glucan-induced trained immunity through blocking mTOR signaling, leading to long-lasting anti-inflammatory activities by IL-38 [75].

All in all, according to the controversial anti-inflammatory effects of IL-38, further investigations are needed for more crucial roles and more efficacious targeted therapies [76].

##### IL-1R1

IL-1R1 is related to IL-1α, IL-1β, and is produced in many cells that activate IL-1R3 following binding of the mentioned IL-1 cytokines [54]. Since IL-38 has 41% homology with IL-1Ra (an IL-1R1A), this cytokine also acts as an IL-1R1 inhibitor [54,62]. The affinity of IL-38 for binding to IL-1R1 is less than two other receptors, which is explained in the next subsections [54].

##### IL-36R

IL-36R is related to cytokines IL-1α, β, γ, and members of the IL-36 subfamily [54]. After the binding of IL-36 subfamily members to IL-36R and IL-1RAPL1, activation of the NF-κB and MAPKs signaling pathways occurs (Figure 5) [54,77]. Since IL-38 has 43% homology in the sequence with IL-36RA, both of them have similar antagonistic effects on IL-36 [54,59,78,79]. It has been demonstrated that the whole length of cDNA in IL-38 is 459 base pairs (bp), encoding the product of 152 amino acids [80].

This receptor is widely expressed in human fibroblast cells and keratinocytes with various patterns of expression in mice and humans [54]. Murine macrophages can express this receptor continuously, whereas, the situation of IL-36R expression in humans is not the same [54]. Additionally, it has been reported that IL-38 is unable to show its immunobiological effects due to the lack of IL-36R in THP1 cells [27,54].

##### IL-1 Receptor Accessory Protein-like 1 (IL-1RAPL1)

IL-1RAPL1 is extensively expressed in the brain tissues and there is no evidence of the expression of IL-1RAPL1 in the lymph node, spleen, and bone marrow [54]. MORA et al. reported that IL-38 was also present on the macrophages and was more expressed in contact with the apoptotic conditioned media. They reported that binding of IL-38 to the receptors on the surface of the macrophages eventuated to inhibit the secretion of IL-6. This result supports the hypothesis that cleaved forms of IL-38 can decrease IL-6 expression by binding to IL-1RAPL1, whereas, full-length forms of IL-38 have the potential to increase it (Figure 5) [54,81].

#### 3.1.3. Inhibitory Properties of IL-38

As we reviewed earlier, on one hand, IL-38 binds to the IL-36R and provides inhibitory effects similar to IL-36RA including anti-inflammatory effects on PBMCs [82]. On the other hand, the pro-inflammatory effects of IL-38 on the DCs are provided by increasing the production of IL-6 [57,82].

In several in vitro studies, it has been reported that IL-38 “directly” inhibits the production of cytokines from THP1 cells. IL-38, only at low doses (10 ng/mL), inhibits the production of IL-17 and IL-22 cytokines, rather than high doses (1 μg/mL). Similar to IL-36RA, IL-38 inhibits the production of IL-8, leading to inhibition of MAPK/NF-κB signaling in keratinocytes (with 30% less potency than IL-36Ra). Truncated and purified forms of IL-38 inhibit IL-36γ-induced secretion of IL-8. Additionally, IL-38 can reduce the pro-inflammatory effects of IL-6, monocyte chemoattractant protein-1 (MCP-1), C–C motif chemokine ligand 2 (CCL2), and A proliferation-inducing ligand (APRIL) up to 30 times [23,27,32,36,41,69,81,82,83,84,85,86,87].

In “indirect” processes, during lipopolysaccharide (LPS) stimulation, IL-38 diminishes Toll-like receptor-4 (TLR-4)-mediated inflammation through decreasing levels of IL-6 and interleukin-23 (IL-23) in THP 1 cells or primary M1 macrophages. Therefore, if targeting TLR signaling in this “indirect” pathway is recruited by an “IL-38-mediated targeting”, it can be considered as a promising therapeutic approach in inflamed skin. However, the signaling pathway of IL-38 still requires further investigation [81,88,89].

Results from several studies have reported that the concentration of IL-38 can be a determining factor affecting the intensity of immunomodulatory properties by IL-38 [82]. As a matter of fact, in high concentrations of IL-38, there is a decreasing trend for its inhibitory effects. This point indicates a relative anthropometric function for IL-38. To explain this more clearly, there is a hypothetical inhibitory curve that IL-38 may use at low concentrations, but this crosstalk has not been scientifically detected yet [82].

### 3.2. Role of IL-38 in Immune-Mediated Diseases

Over the past few years, the tendency to investigate the immunobiological roles of IL-38 in immune-mediated diseases has matured [90], which is shown by the schematic presentation in Table 1.

#### 3.2.1. Role of IL-38 in Autoimmune Diseases

##### Rheumatoid Arthritis (RA)

RA is a very common autoimmune disease with long-term chronic inflammation of the synovium, mainly causing swollen and painful joints in the wrist and hands. RA is commonly manifested by a systemic inflammation in the hands, fingers, knees, ankles, elbows, hips, and shoulders. These clinical manifestations are related to the cellular interactions between resident cells (fibroblast-like synoviocytes (FLS)) and immune cells involved in the innate and adaptive immune system [112]. IL-38 is possibly induced by synovial fibroblasts, monocytes, and macrophages in RA, and it can diminish the immunobiological functions of IL-36 agonists in the pathogenesis of RA. IL-36 agonists, IL-36RA and IL-38, were induced in thee synovium of the majority of patients with RA, and the total balance in the pathogenesis of RA was attributed to these potential antagonists. These are produced by many different cell types such as myeloid and plasma cells, and their expression correlates with macrophage-colony stimulating factor (M-CSF), CCL3, and CCL4. Other cells such as endothelial cells, fibroblasts, and enterocytes could also participate in RA pathogenesis [12,54,85].

Consequently, on one hand, overexpression of IL-38 induces anti-inflammatory effects in mice with RA as well as human macrophages in vitro. In a clinical study, a significant decrease in the filtration of macrophages as well as expression of TH17 cytokines were reported after an articular injection of IL-38-encoding adeno-associated virus (AAV). Additionally, there was a significant decrease in the clinical scoring of inflammation in collagen-induced arthritis (CIA) and serum transfer-induced arthritis (STIA) model mice [12,113]. On the other hand, IL-23, whose expression is normally reduced by IL-38, is highly expressed by DCs. Therefore, in this study, it was reported that IL-38 could target DCs as well as macrophages. At the same time, it did not affect cartilage and bone destruction. Interestingly, over-expression of IL-38 may lead to an augmented expression of osteogeneic factors, inhibited neovascularization, and improved damage in cartilage of RA rabbits or murine models of clinical trials. Hence, IL-38 might be promising to consider as a potential seroimmunobiomarker, being targeted for the regenerative medicine purposes of RA [113,114,115,116,117].

Increased levels of the IL-38 cytokine in mouse models of RA has a positive correlation with synovial fluid levels of IL-1β and a negative correlation with IL-17 responses as well as with TNF-α production. IL-38 has the potential to suppress IL-17 and IL-22 secretion, similar to IL-36Ra. Upsurged levels of the IL-1 superfamily and IL-36 subfamily members depict a negative feedback for the inhibition of inflammatory responses in RA [24,63]. Additionally, immunomodulatory properties of IL-38 in the pathogenesis of RA likely reduced the filtration of macrophages into the synovium, and the reduced TH17 production should not be underestimated [12,54,103].

In CIA immunopathogenesis, it is noteworthy that the increase in IL-36/IL-36Ra occurs in the initial phases, whereas the elevation of articular IL-38 is postponed and accurately will occur at the later phases. The excessive expression of IL-38 attenuates the severity of clinical manifestations in CIA and STIA types of RA, and improves the clinical scores of the disease through the reduction in TH17 cytokine patterns. Conversely, excessive expression of IL-38 led to more severe clinical manifestations in antigen-induced arthritis (AIA). Additionally, overexpression of IL-38 did not indicate any significant effects on bone erosion (contrary to IL-1RA) [63,69,81,113,118].

To sum up, overexpression of IL-38 in the synovial membrane and sera leads to lessened functions of macrophages, TNF-α, IL-6, IL-10, TH17 cytokines (IL-17, IL-22), C-X-C motif chemokine ligand 1 (CXCL1), and CXCL8 expression [68,119]. Correspondingly, it was demonstrated that IL-38 showed suppressive impacts on LPS-mediated TLR4 secretion, leading to an attenuated inflammation in patients with RA through inhibition of the activation of NF-κB [120]. Therefore, if we consider both IL-36RA and IL-38 cytokines as potential IL-36 antagonists, only a minor subpopulation of patients with RA (17–29%) have an elevated agonist/antagonist ratio [85,121].

Takenaka et al. reported higher serum and synovial levels of the IL-38 protein in patients with RA as well as lower concentrations of IL-38 in osteoarthritis (OA) patients and normal subjects. Serum levels of IL-38 in patients with RA did not correlate with their disease activity, treatment, or disease duration [122]. In another study, Wang-Dong Xu et al. found that there was a direct correlation between IL-38 expression and disease activity in RA patients. Along with exacerbation in the inflammatory conditions, there was a significant reduction in the levels of IL-38 expression. Hence, this cytokine was identified as an appropriate and promising seroimmunobiomarker for the diagnosis of RA and determination of RA severity [113,115,123].

In another newly conducted comparative study by Lifeng Jiang et al. [124], they reported promising findings on the association of IL-38 expression in the chondrocytes of patients with OA. Assessment of IL-38 was conducted on serum and synovial fluid of 75 patients with OA, who had undergone joint replacement before the assay, and 25 age- and sex-matched healthy persons as a control group. They reported remarkably elevated serum and synovial fluid levels of IL-38 in patients with OA when compared to the control group. These findings were proven by an attenuated expression of several pro-inflammatory cytokines and were positively correlated with early disease activity. Altogether, they reported that IL-38 could promisingly serve as a novel immunotherapeutic target, and a screening seroimmunobiomarker for patients with OA [124]. Results of this study are in accordance with another study aimed at evaluating the anti-inflammatory properties of IL–38 in murine models of arthritis and systemic inflammation [125]. In this study, inhibitory features of IL-38 in streptococcal cell wall (SCW)-induced arthritis and monosodium urate (MSU) crystal-induced arthritis were recognized by a reduction in joint swelling, inflammatory cell influx, and synovial levels of several pro-inflammatory cytokines, clearly indicating the potential abilities of IL-38 as an acceptable seroimmunobiomarker and immunotherapeutic target for patients with arthritis [125]. Additionally, Mitra Abbasifard et al. reported higher expression of IL-38 in patients with severe OA, which were approved by WOMAC scores > 40, VAS scores > 5, and BMI < 25 indices in 23 newly-diagnosed patients with OA compared to 22 sex- and age-matched healthy persons as a control group [126].

##### Systemic Lupus Erythematosus (SLE)

Multiple genetic predispositions and environmental factors contribute to aggravated pathogenesis of SLE as a multi-systemic autoimmune disease. In patients with SLE, a wide range of organs and tissues of the host (including skin, joints, blood, CNS, and kidneys) are clinically affected. Tissue damage and disruption of their normal functions are direct consequences of long-time inflammation in patients with SLE. Subsequently, the main goal of SLE therapy will be a reduction in inflammation in the immunopathogenesis of the diseases [118]. There are variations caused by complex single-nucleotide polymorphisms (SNPs) in the locus of the IL-1 superfamily, which are associated with a higher activity of IL-1 superfamily members (especially IL-1A activity), altering the susceptibility of individuals to SLE development [127].

In a C57/BL6 mouse model-based study, the immunobiological effect of IL-38 injection in gouty arthritis was investigated through the administration of albumin-opsonized MSU crystals. It was reported that intraperitoneal injection of 1 µg recombinant IL-38 had alleviated proteinuria, skin lesions, and nephritis two hours before the induction of gouty arthritis. Additionally, injection of 1 µg recombinant IL-38 also reduced the serum levels of IL-17 and IL-22 cytokines [63,81]. Recombinant IL-38 induces downregulatory impacts on the secretion of pro-inflammatory cytokines (IL-17, IL-22, and IL-36γ), and increases the expression levels of IL-6, CCL2, and APRIL. These findings, all in all, indicate the anti-inflammatory activities of IL-38 on disease activity and organ involvement in a mouse model of gouty arthritis [85].

Results of another study showed that concentration of serum IL-38 in patients with SLE was significantly higher compared to the healthy controls [63]. On the other hand, in patients with active SLE, the expression levels of IL-38 were higher than the ones in patients with inactive SLE. Hence, the risk assessment for renal and CNS involvement is directly related to the level of IL-38 [55,63,85,118,128].

In their experimental study, Takeuchi et al. evaluated the serum levels of IL-38 in 19 patients with SLE (juvenile SLE) through double-sandwich enzyme linked immunosorbent assay (ELISA) and IHC (using a polyclonal anti-IL-38 antibody). They reported undetectable serum levels of IL-38 in almost all cases throughout the disease course at the diagnosis time before treatment (except one case). They suggested that IL-38 may not be a sensitive disease-activity (disease-specific) biomarker in pediatrics with SLE. In addition, they suggested further investigation into the expression levels of IL-38 in larger samples of patients with juvenile SLE [129]. Another study introduced IL-38 as the first mediator, being associated with disease severity, renal, and CNS involvement in patients with SLE [130].

Chu et al. reported that intravenous administration of murine recombinant IL-38 into specific kinds of mice can ameliorate skin inflammation and nephritis in mice with SLE, probably via suppressing the secretion of inflammatory cytokines such as IL-17 and IL-22. They suggest that IL-38 shows therapeutic potential for the recovery of skin and kidney involvement in mice with SLE [131].

##### Psoriasis

As a common chronic inflammatory disorder, psoriasis is accompanied by an uncontrolled proliferation of keratinocytes, several chemokines, and excessive cascades of DCs infiltration, leading to a wide range of clinical manifestations such as dermatologic lesions, acanthosis, and cardiovascular comorbidities [5,38]. Severe clinical manifestations in psoriatic patients emerge due to an aberrant expression of inflammatory agents such as TNF-α, IL-2, IL-6, IL-12/IL-23p40, IL-17A, IL-17/IL-23, and interferon-gamma (IFN-γ) cytokines. Correspondingly, a wide array of novel monoclonal antibodies (Secukinumab, Briakinumab, Brodalumab) are going to be therapeutically used in clinical trials for psoriasis according to the involved cytokines in the immunopathogenesis of psoriasis [5,72].

In a study accomplished by Han Y et al., they reported that in IL-38 KO mice, the process of disease resolution is postponed due to an aberrant secretion of IL-17, orchestrating inflammation in the microenvironment of the diseases. Immune cells such as γδ T cells are responsible for IL-17 cytokine production. There is a direct relationship between the activation of γδ T cells and upregulation of IL-1RAPL1. Additionally, there are diminished levels of IL-17 secretion and reduced inflammation by activated γδ T cells in psoriatic IL-1RAPL1 KO mice. Additionally, clinical administration of mature IL-38 or γδ T cell-receptor-blocking antibodies leads to an accelerated amelioration of the disease. In another study, higher expression levels of IL-38 were detected in pustular psoriasis in comparison with psoriasis vulgaris and healthy controls [72,81,132]. Since IL-38 is reduced in differentiated keratinocytes, the loss of IL-38 in the epidermal layers can play an important role in the severity of pathogenesis in psoriatic patients. Additionally, constitutive overexpression of IL-38 decreases the proliferation and viability of normal human keratinocytes (NHKs), and is able to increase cell mortality, but due to the lack of sufficient experiments, it has not been fully understood yet [54,133,134]. Mast cells play a contributing role in the immunobiology of IL-38, releasing IL-1 and stimulating the macrophages to express IL-36 with its pro-inflammatory activities. Then, IL-38 and IL-36RA bind to IL-36R and inhibit inflammation in the psoriasis microenvironment, holding promising immune gene-based therapeutic capabilities of IL-38 and IL-36RA for psoriatic patients [135].

IL-38 polymorphism is associated with psoriatic arthritis, suggesting possible roles of a higher expression levels of IL-38 in the lesional skin and exacerbated immunopathogenesis of this inflammatory skin disease. Conversely, in vitro addition of IL-38 in PBMC culture could inhibit the production of IL-22 and IL-17A, presenting IL-38 as an agent involved in the regulation of expression levels of IL-17 [61,136,137]. Although the exact role of IL-38 is not completely elucidated, the correlations between SNPs in the regions encoding IL-38 and susceptibility to psoriatic arthritis are thought to have contributed [138].

Boutet et al., in their mice model-based study, reported that IL-36RA is induced in psoriatic skin, whereas IL-38 is reduced. Considering both IL-36RA and IL-38 as potential antagonists of IL-36, the majority of patients with psoriasis revealed an elevated agonist/antagonist ratio, proposing key roles of IL-36 cytokines in psoriasis and their correlation with disease severity. In contrast, a reduction in IL-38 could have an important role in the immunopathogenesis of psoriasis through activated IL-36 agonists with pro-inflammatory properties. Hence, IL-38 can be promisingly considered as a prognostic seroimmunobiomarker for psoriasis [61,139].

Jennifer Palomo et al. measured the inhibitory effects of IL-38 on IL-36R in patients with psoriasis based on the fact that IL-36R deficiency can decrease IMQ-induced skin inflammation in psoriatic mice. Interestingly, their results showed no inhibitory effects of IL-38 on IL-36R in patients with psoriasis [140]. Additionally, according to the results of another study, the inhibitory effects of IL-38 on inflammatory functions of IL-36 in the skin have not been proven [64].

##### Atopic Dermatitis (AD)

As a chronic and itchy skin disorder, AD is immunologically recognized with an exacerbated inflammatory dermatosis, and intensified levels of TH2 cytokines. Although some foods and environmental factors affect pathogenesis of AD, the real etiopathogenesis of AD is incompletely understood. Khattab et al., in their case-control study, surveyed the serum levels of IL-38 in patients suffering from mild/moderate/severe atopic eczema compared to the age- and sex-matched healthy controls. Results of ELISA revealed higher expression levels of IL-38 in patients with atopic eczema. In addition, they reported a significant correlation between serum levels of IL-38, disease severity, eosinophilia, and IgE levels. Altogether, elevated expression levels of IL-38 in the sera of patients with AD can potentiate the specificity of eosinophilic count and IgE as inducer agents of TH2 responses, presenting IL-38 as a prognostic seroimmunobiomarker for disease severity [105].

In another most recently-published article, Lauritano et al. introduced allergic contact dermatitis as a provoker agent for AD immunopathogenesis. As is clear, involvement of allergens and secretion of a majority of pro-inflammatory mediators (e.g., IL-1 superfamily members) contribute to the aggravation of the pro-inflammatory situation in allergic contact dermatitis [141]. They reported that there are chains of anti-inflammatory activities related to IL-38, mainly including lessened allergic inflammation in the skin. Several immunobiological cascade processes by IL-38 are responsible for this reduced allergic inflammation, and suppressed activities of IL-36, introducing IL-38 as an immunotherapeutic target for allergic contact dermatitis. Altogether, the binding of mature IL-38 to IL-36R (IL-1R6) orchestrates the induction of inhibitory effects on the production of mast cell-derived proteases, prevention from the formation of biologically-active IL-36, expansion of common proteases for IL-38 maturation, induction of inhibitory effects on the secretion of IL-2, IL-17 cytokines, and inhibited production of IL-1, especially in macrophages [141].

##### Kawasaki Disease (KD)

KD is one of the inflammatory diseases that is commonly accompanied by vasculitis in children, increasing the risk of cardiovascular diseases (CVDs), especially when it remains unresolved or mistreated. Environmental stimuli are also significant determinants in the clinical symptoms and manifestations of KD [79].

According to the immunological patterns of KD and major anti-inflammatory properties of IL-38, it was demonstrated that in children with KD, the levels of IL-38 were higher in patients with an acute phase of KD compared to the healthy control group. These results reveal the immunobiological functions of IL-38, contributing to the inflammatory process of KD [121].

##### Gouty Arthritis

As one of the prototypes of IL-1β driven auto-inflammatory diseases, gouty arthritis clinically results from an excessive formation of acid uric crystals, especially in the joints of the toes. In this disease, neutrophils and other involved immune cells are collected in the sedimentation site following the deposition of MSU crystals in the joints [119]. A narrow range of studies have been conducted to investigate the role of inflammatory factors and cytokines in the etiopathogenesis of this disease [142].

In an experimental study on a mouse model of gouty arthritis, it was hypothesized that intraperitoneal administration of recombinant IL-38 leads to a reduction in inflammation in the joints and swelling. These profound alterations in the clinical manifestations of the disease are due to reduced secretion of IL-1β and IL-6 in the synovial membrane. These results indicate that IL-38 might be used as an acceptable cytokine-based targeted immunotherapeutic approach for gouty arthritis [63].

##### Sjögren Syndrome (SS)

SS is a chronic systemic immune-mediated inflammatory disorder mainly characterized by abnormal functions of exocrine glands, dryness, or irritability of the eyes, sensation of foreign bodies in the eyes, corneal scarring, dry mouth, and oral ulcers. The most recently known immune axis involved in the immunopathogenesis of SS is the IL-23/IL-17/IL-22 axis [143,144]. Not only do these patients suffer from poor lifestyle due to the abnormal clinical symptoms, but patients with SS are also prone to a shortened life expectancy by increasing the risk of non-Hodgkin’s lymphoma [145].

In a study investigating the salivary gland levels of IL-38 in patients with primary SS (pSS), they reported higher levels of IL-38 in patients with pSS compared to the control group [58].

In another study conducted by Ciccia F et al., it was reported that there was an upregulation in minor salivary gland and serum levels of IL-36α in a minority of patients with pSS, leading to a more severe disease activity. Additionally, in their minor salivary glands, there was a down-regulation in the expression levels of IL-36RA, and an upregulation in the expression levels of IL-38, aimed at counteracting the imbalanced activation of IL-36 [144]. The latest molecular-based study conducted on the role of IL-38 in SS introduced IL-38 as an inhibitory agent for the secretion of chemokines involved in the TH17 signaling pathway, preventing the exacerbation of immunopathophysiology involved in pSS, and acting as a promising targeted immunotherapeutic-based treatment for patients with pSS [146].

##### Crohn’s Disease (CD)

As a chronic inflammatory disease, the gastrointestinal tract is attacked by the host immune system in CD. Many studies have been conducted to understand the mechanisms of the inflammatory pathways in CD and significant results have been obtained [147]. Mice with colitis and patients with CD showed weakly augmented levels of IL-36α and IL-36γ. In a subgroup of patients with CD, there was a correlation between the expression levels of IL-36α, IL-36γ, IL-38, and IL-17A. Hence, these results confirm a role for these cytokines in the immunopathogenesis of CD, which can be used as prognostic or diagnostic approaches for patients with CD [148].

In mouse and human inflamed colon, increased expression levels of IL-36α, IL-36γ, and IL-38 have been experimentally observed. Moreover, in an IL-36R-deficient mice, colitis development at early time points as well as defective recovery from mucosal injury were diminished. These results suggest major roles for IL-36γ in colonic inflammation and its resolution. Further investigations are necessary to precisely clarify the immunobiological roles of other members of the IL-36 subfamily and IL-38 in the immunopathogenesis of CD [149].

In another study aimed at the investigation of IL-38 levels in patients with RA and CD, it was reported that induced production of IL-38 (possibly by synovial fibroblasts, monocytes, and macrophages) could limit the immunobiological impacts of IL-36 agonists in the immunopathogenesis of RA and CD. Future studies will certainly unravel more clues on the role of IL-38 in the immunopathophisiology of chronic inflammatory diseases such as CD. Furthermore, there is an urgent need to precisely decipher the IL-38 antagonistic potencies against members of the IL-36 subfamily or other triggered inflammatory agents through in vitro and in vivo studies. In the colon of the majority of patients with CD, IL-36α, IL-36γ, and IL-38 were induced at low levels, potentiating IL-38 as an antagonist. These are produced by different cell types such as myeloid cells and plasma cells, and their expression correlates with TH17 cytokines [139].

In an original study designed by Fonseca Camarillo et al., they characterized tissue expression of IL-38 and IL-36Ra, and their producing cells in patients with ulcerative colitis (UC), CD, and other patients with remission of IBD. Colonic tissue levels of IL-38 were increased in patients with inactive UC compared to active UC and control groups. Correspondingly, in patients with active CD, plasmacytoid DCs (pDCs) showed a higher number on sub-mucosa, muscles, and adventitia compared to patients with active UC and non-inflamed control tissue [148].

In another cross-sectional comparative study, *Fonseca Camarillo* et al. evaluated the expression levels of IL-36*α*, IL-36*β*, IL-36*γ*, IL-36Ra, and IL-38 genes, their producing cells, and their correlation with clinical activity in IBD and non-inflamed non-IBD persons (as a control group) by reverse transcriptase-polymerase chain reaction (RT-PCT) and double-staining IHC techniques. In comparison with patients with CD and the control group, there was an increase in the colonic mucosa levels of IL-36 subfamily members in patients with active UC (except than IL-38) [150], whereas, laboratory data from real-time RT-PCR depicted an increase in the tissue levels of IL-38 patients with inactive UC compared to patients with active IBD and non-inflamed non-IBD control groups. In addition, overexpression of IL-36*α*, IL-36*β*, IL-36*γ*, IL-36RA, and IL-38 was reported in intestinal epithelial cells, macrophages, and CD8^+^ T cells of patients with active IBD compared to the non-inflamed non-IBD control group [150]. Altogether, elevated secretion of IL-38 from immune and non-immune cells in patients with active IBD introduces IL-38 as a practical screening and immunotherapeutic seroimmunobiomarker for gut inflammation [150].

Yuxia Zhao et al. [151] designed their comparative study to investigate the expression levels and molecular mechanisms of IL-38 in 115 patients with CD, 67 patients with UC as well as a control group (40 people) admitted to the referral *Wuhan Children*’*s Hospital*. Several parameters such as C-Reactive protein (CRP), erythrocyte sedimentation rate (ESR), phosphorylated signal transduction and activator of transcription 3 (p-STAT3), IL-38, and NF-κB were measured in the intestinal mucosa of the studied population through IHC staining [151]. Crohn’s Disease Activity Index (CDAI) for patients with CD, and Mayo score system for patients with UC were also assessed [151]. The expression levels of IL-38 were quantitatively assessed by ELISA. For their mice-based molecular assessments, they established a dextran sulfate sodium (DSS) model for the induction of IBD in IL-38-C57BL/6 transgenic mice. Inflammatory markers and cells in the intestinal mucosa of the wildtype and IL-38-C57BL/6 transgenic mice were assessed by IHC and flowcytometry techniques, respectively [151]. Results of this study reported lower intestinal mucosa levels of IL-38 in both patient groups when compared to the control group (certified by IHC and ELISA), whereas decreased expression levels of mentioned inflammatory markers were found in the control group when compared to both case groups. There was a reverse correlation between disease activity, and expression levels of CRP, ESR, and IL-38. Results acquired from their mice-based study, showed similarity with the ones obtained from the patients, indicating the therapeutic potentials of IL-38 in the immunopathogenesis of IBD by a reduction in inflammation through the inhibitory effects on p-STAT3 and NF-κB [151].

##### HLA-B27-Associated Anterior Uveitis and Idiopathic Anterior Uveitis (IAU)

IAU is an autoimmune systematic inflammatory disease, affecting anterior segment, iris, and ciliary body of the eye. Pathogenically autoreactive TH17 cells as well as increasing levels of IL-38 and IL-37 are of high prominence for investigation as diagnostic or therapeutic tools for patients with IAU [152].

#### 3.2.2. Role of IL-38 in Inflammatory Disorders

##### Inflammation-Induced Corneal Neovascularization

To study the immunobiological effects of IL-38 on corneal neovascularization, an alkali-induced corneal neovascularization mouse model has been used. It has been interrogated that if IL-38 is topically administrated in the injured cornea of the mentioned model, inflammation-induced angiogenesis is diminished. IL-38 is capable of attenuating thee secretion of IL-1β, IL-6, IL-8, and TNF-α cytokines and reducing angiogenesis-related activities (e.g., proliferation, migration, and tube formation of human retinal endothelial cells) [153].

##### Hidradenitis Suppurativa (HS)

HS is chronic inflammatory disease presented with recurrent painful inflamed lesions due to the inflammatory processes in the terminal hair follicles. In a study, researchers aimed to evaluate the lesional and perilesional skin levels of the IL-36 subfamily members, IL-37 and IL-38 in patients with HS compared to the healthy controls [154]. Compared to healthy controls, overexpression of IL-38 and reduced titration of IL-38 were reported in the perilesional skin and lesional skin samples of patients with HS, respectively. These results have been confirmed by real-time RT-PCR and IHC in these samples. Additionally, here, IL-38 showed anti-inflammatory properties by the suppression of IL-17 and IL-22 cytokines [55,154].

##### Chronic Primary Angle Closure Glaucoma (CPACG)

As one of the worldwide leading causes of irreversible blindness, CPACG is mainly reported among female populations of Asian nations with a loss of visual functions [155,156,157]. It has been demonstrated that there is an upregulated immune-mediated inflammation in acute primary angle closure glaucoma (APACG) and CPACG. For instance, there were increased levels of IL-6, IL-8, granulocyte-colony stimulating factor (G-CSF), MCP-1, and IP-10 in the aqueous humor of eyes or sera of patients with CPACG and APACG. In addition, in an experimental study conducted by Jin-ling Zhang et al. (2019), there were increased levels of IL-38 in the aqueous humor of patients with CPACG in comparison with age-related cataract (ARC) samples. In addition, a significant correlation was reported between IL-38 levels and mean deviation of visual field (MDVF), proposing IL-38 as a critical factor in the immunopathophisiology of CPACG, even in the absence of a clinically confirmed inflammation. Hence, it highlights the prognostic values of IL-38 in patients with CPACG [157].

##### Retinal Ischemia and Proliferative Vascular Diseases

Damage from ischemic vascular diseases in the retina vary from a slow degradation of vision (due to the permanent ischemic retina) to a completely destroyed vision (due to proliferative vascular disease). Oxygen-induced retinopathy (OIR) is a mouse model for studying retinal ischemia and proliferative vascular diseases in the retinal vasculature, demonstrating pivotal roles of several angiogenic factors and inflammatory cells in the aggravation of vascular immunopathogenesis [158].

Results acquired from a murine model-based study reported significant anti-angiogenic functions in the retina of the hyperoxia mice due to the injection of IL-38. This injection led to an induction in the tube formation of vascular endothelial cells via the role of vascular endothelial cell growth factor (VEGF) and reduced severity [159].

#### 3.2.3. Role of IL-38 in Metabolic Disorders

##### Type2 Diabetes Mellitus (T2DM) and Obesity

Obesity is clinically accompanied by chronic inflammatory responses. The main mechanism accounting for obesity is a disturbed balance between receiving and consuming energy. This imbalance may be due to the decreased physical activities or increased food intake [160,161], and insulin resistance caused by T2DM [162]. T2DM is a metabolic disorder in which several agents such as genetic factors, immunological agents, and disturbed immune system play indispensable roles in aggravating the etiopathology of T2DM. As a prime instance, overexpression of IL-6 can lead to complications such as dyslipidemia, which in turn, can cause obesity [163].

In a study, they found that SNPs in the IL-38 gene were associated with high serum levels of CRP in patients with obesity and T2DM. Based on this evidence, IL-38 is likely to play an important role in the inflammation involved in the immunopathogenesis of obesity and T2DM [164]. It has been demonstrated that there is a positive correlation between overexpression of IL-38 and diabetic nephropathy in patients with T2DM [165].

In a newly conducted experimental study by Keye xu et al., the expression levels of IL-38 were assessed in a mouse model of high-fat diet-induced obesity after hydrodynamic-based IL-38 gene delivery. Results of this study reported the anti-inflammatory effects of IL-38 through the inhibition of IL-1β, IL-6, and MCP-1, and a reduction in liver fat content due to reduced adipose tissues [161]. Results of this study are in accordance with another in vitro study that investigated the biological effects of IL-38 on adipogenesis and the secretion of inflammatory cytokines from adipocytes [166]. IL-38 also decreased the number of lipid droplets in adipose precursor 3T3-L1 cells, inhibited the secretion of IL-1β, IL-6, and MCP-1, and reduced the number of differentiated adipocytes. These studies promisingly indicate IL-38 as an immunotherapeutic target for patients with obesity [166].

Additionally, in the most recently accomplished comparative study by Ying Liu et al., the expression levels of IL-38 and underlying inflammatory mechanisms were quantitively investigated in patients with T2DM and the ones with no abnormality in glucose metabolism (as a control group). Expression levels of IL-38 were statistically related to T2DM and insulin resistance [167]. Afterward, they divided the study population into two categories including the ones resistant and sensitive to insulin therapy. Higher expression levels of IL-38 were reported in children sensitive to insulin therapy. In another part of this study, the results from the mouse model comprised the suppressed expression levels of IL-36 and improved clinical outcomes as inhibited development of T2DM due to the overexpressed levels of IL-38 [167]. Results of this study were in accordance with other study conducted on 21 Iraqi patients with T2DM, and age- and ethnic background-matched people as a control group. Lower concentrations of IL-38 were presented with an increased risk of insulin resistance, obesity, and T2DM. Altogether, it seems that IL-38 can serve as an immunotherapeutic seroimmunobiomarker for cases with obesity and as a prognostic marker for the determination of susceptibility to T2DM [168].

##### Pregnancy and Gestational Diabetes Mellitus (GDM)

As a form of diabetes, GDM appears for the first time after the first trimester of pregnancy without any predisposition history of type 1 diabetes mellitus (T1DM) or T2DM [169]. Results of a study showed a significant increase in the expression levels of IL-38 at the arteries and veins of the umbilical cord and placenta in patients with GDM. These data indicate that IL-38 plays an indispensable role in inhibiting localized inflammation in diabetic patients, although the underlying mechanisms for this role are not completely known as yet. However, the anti-inflammatory effects of IL-38 may be useful for patients with controlled diabetes. There is a direct relationship between fasting blood sugar (FBS) and IL-38, reflecting the immunobiological roles of IL-38 in the development of GDM [133].

Recently, Southcombe et al. reported that IL-38 is expressed by the human placenta. They investigated the serum levels of IL-38 in women with normal pregnancy and with pre-eclampsia and determined IL-38 localization in the placenta and its release. In this study, they found no notable differences in the expression levels of IL-38 between women with pre-eclamptic and normal placentas [170].

##### Hyperlipidemia

Hyperlipidemia, as increased levels of cholesterol or triglycerides or chylomicrons, encircles several disorders that can be clinically manifested as glaucoma, ocular hypertension, and intraocular pressure. Hyperlipidemia can emerge familiarly or be caused by other predisposition diseases such as diabetes [171].

In their study, Ning Young et al. investigated the immunological roles of IL-38 in the development of hyperlipidemia and reported higher blood levels of the IL-38 protein in patients with hyperlipidemia compared to the control group. In addition, expression levels of IL-38 mRNA were higher in PBMC samples of the patients with hyperlipidemia in comparison with the control group. In this study, they found that in patients with hyperlipidemia who responded well to atorvastatin, the expression levels of IL-38 were higher than those who were resistant to this drug. Results of the in vitro investigations showed the inhibitory effects of IL-38 on the expression levels of IL-1β, IL-6, and CRP in PBMCs. In addition, they reported increased expression levels of IL-38 and inhibited the deterioration of hyperlipidemia. Finally, they concluded that IL-38 may be a promising and effective targeted immunotherapeutic for hyperlipidemia [172].

#### 3.2.4. Role of IL-38 in CVDs

##### Myocardial Infarction (MI)

MI is considered as one of the direct consequences of consecutive hypo-perfusion in myocardial cells. Following MI, the cellular immune system attempts to regenerate the damaged tissues by producing various cytokines and increasing the levels of CRP [173]. Polymorphisms in IL-38 were associated with CRP concentrations in the sera of patients with stroke. In addition, IL-38 mRNA was found in human atheromatous plaques of patients with coronary artery disease (CADs) [32]. It has been reported that reperfusion-based strategies could decrease the above parameters, and the IL-38 levels were positively correlated with CRP, cardiac troponin I (cTnI), and NT-proB-type natriuretic peptide (NTproBNP) whereas the expression levels of IL-38 were in a weakly negative correlation with the left ventricular ejection fraction (LVEF) [32,83]. Therefore, it seems that IL-38 might be an efficient prognostic seroimmunobiomarker for the expansion of infarction, cardiac rupture, severity of MI, and risk assessment of mortality in patients with CVDs [174].

In another study, plasma levels and gene expression of IL-38 in PBMC samples were significantly increased in ST-elevated myocardial infarction (STEMI) patients in a time-dependent manner compared to the control group (peaked at 24 h). In addition, plasma levels of IL-38 were dramatically reduced in patients with hyper-perfusion who had undergone treatments compared with the control group. Moreover, no significant differences in IL-38 level were reported between diabetic and non-diabetic STEMI patients. In addition, the assessment of plasma IL-38 levels could be a criterion in monitoring the response to judging the successfulness of percutaneous coronary interventions (PCI) in patients with STEMI as a seroimmunobiomarker for monitoring treatment [174].

In another newly published study, Yuzhen Wei et al. reported increased expression levels of IL-38 in the C57BL/6 mouse model of MI. In addition, after the injection of recombinant IL-38, there was ventricular remodeling post-MI, restricted inflammatory responses, attenuated myocardial injury, and diminished myocardial fibrosis through affecting (altering) the phenotype of DCs. Altogether, IL-38 could be considered as a promising and potential immunotherapeutic target for MI [175].

Another newly conducted study conducted by Dennis Marinus De Graaf et al. [176] aimed at investigating the association of plasma concentration of IL-38 in over-weight patients with CVDs and 288 healthy European participants. Results of this study reported lower expression levels of IL-38 in CD19^+^ B cells when compared to cellular immunity-associated expression. Hence, they concluded that patients with B cell deficiency had decreased expression levels of IL-38 and increased systemic inflammation, accompanied by increased risk for CVDs and other metabolic disorders as direct consequences [176].

#### 3.2.5. IL-38 and Cancer Co-Relation

There is a correlation between cancer immunopathology and microenvironments with inflammatory factors. For instance, Lv et al. reported that tumor-associated transcriptional factors such as c-Fos, AP-1, c-Jun, and NF-κB could bind to the upstream region of the IL-36RN gene, encoding anti-inflammatory cytokine IL-36RA. This finding may indicate that IL-38 is involved in carcinogenesis and tumor progression [177]. Additionally, increased expression levels of IL-38 in apoptotic cells showed the pro-tumoral behavior of IL-38 [70]. As a matter of fact, on one hand, binding of mature IL-38 to IL-1RAPL1 with high affinity can activate JNK signaling, leading to an indirect induction of tumor-promoting inflammation. Biologically, Mora et al. introduced an IL-38-related apoptotic process as a tolerogenic factor [178]. From functional aspects, precursors of IL-38 can upregulate the expression levels of IL-6 by human macrophages in apoptotic tumor cell conditioned medium (ACM), whereas the truncated forms of IL-38 downregulate expression levels of IL-6 secretion through the attenuation of JNK/AP-1 signaling as a downstream process to IL-1RAPL1. On the other hand, as an antagonist for IL-36R, IL-38 shows inhibitory effects on anti-tumoral activities of IL-36γ [30,89,178].

In the latest clinical study conducted by Feng Wang et al., possible correlations between IL-38 and clinical outcomes of patients with non-small cell lung cancer (NSCLC) (including survival rate, cancer progression, and sensitization to chemotherapy) were investigated. Overexpression of IL-38 suppressed the colony migration and cell proliferation in NSCLC cells and also increased the sensitization to chemotherapy drugs. These findings introduce IL-38 as an efficient immunotherapeutic target as well as a novel prognostic indicator for patients with NSCLC [179].

In a study conducted by Feier Chen et al., they interestingly reported a 95% decrease in the expression levels of IL-38 in colorectal cancer (CRC) tissues compared to non-cancer adjacent tissues [180]. In IHC techniques, stains were diffusely distributed within the cytoplasm and on the cell membrane of the cancerous cells in CRC tissues. They promisingly concluded that IL-38 has a considerable specificity and sensitivity to be introduced as a prognostic seroimmunobiomarker for post-surgery longer survival rate as well as an efficient immunotherapeutic target for patients with CRC. Of note, IL-38 can be considered as a potential therapeutic target for “precision medicine” in patients with CRC according to the tumor size, invasion, metastasis, and position of the tumor for each patient [180]. The results of this study are in accordance with another study that investigated the role of IL-38 and the underlying mechanisms involved in immunopathophysiology in the microenvironment of CRC [181]. They found that there was a significant correlation between IL-38 and prognosis or overall survival rate in the advanced stages of CRC. Additionally, they implied that it is possible for staging and immunotherapeutic roles of IL-38 in “precision medicine”. Hence, the assessment of IL-38 can stop generalizing the results to all of the patients and increase specificity, sensitivity, and efficacy of diagnosis and treatment for patients with CRC [181].

#### 3.2.6. Role of IL-38 in Lung and Respiratory Diseases

##### Lung Fibrosis

Borthwick et al., in 2016, reported that there was no documentation regarding the role of IL-38 in lung fibrosis. Meanwhile, Takada et al., in 2017, published an article investigating the role and expression levels of IL-38 in lung fibrosis [58]. The IL-38 protein was not strongly expressed in normal pulmonary alveolar tissues in all 22 control lungs. In contrast, IL-38 was overexpressed in the lungs of four out of five (80%) patients with acute exacerbation of idiopathic pulmonary fibrosis (IPF) and 100% (10/10) of the patients with drug-induced interstitial lung disease (ILD). IL-38 overexpression was limited to hyperplastic type II pneumocytes, reflecting regenerative changes following diffuse alveolar damages in ILD. Altogether, they declared the contributing roles of IL-38 in acute or chronic inflammation in anti-cancer drug-induced ILD and acute exacerbation of IPF. In other words, IL-38 is of immunopathological significance in both aforesaid respiratory disorders [182].

In another recently conducted study by Zhiwei Xu et al. [183], the association of IL-38 expression and chronic inflammation in a murine-based lung fibrosis model was investigated. They reported that administration of a lentivirus-expressing IL-38 as a transgenic vector to the nasal cavity of a mouse with bleomycin-induced pulmonary fibrosis had led to lowered weight loss and an attenuated pulmonary inflammation. These findings were proved by the reduced production of pro-inflammatory cytokines and increased the secretion of IL-1RA in the lungs by IL-38. Therefore, they believed that these data indicate the anti-inflammatory activities of IL-38 in lung tissues, suggesting the promising therapeutic roles of IL-38 for patients with pulmonary fibrosis [183].

Yun-Hui SUN et al. [184] assessed the correlation between the expression levels of IL-38 and MIP-2 in a bleomycin/dexamethasone-induced pulmonary fibrosis model, and compared their results with the normal group (saline group) [184]. They used Hematoxylin and Eosin (H&E) staining, ELISA, and RT-PCR for the investigation of histopathological changes and the expression levels of IL-38 and MIP-2, respectively. Increased inflammation accompanied by MIP-2 and decreased expression of IL-38 were reported in both case groups (bleomycin and dexamethasone-receiving groups). Due to increased levels of IL-38 and decreased levels of MIP-2 expression in dexamethasone-receiving mice in comparison with bleomycin-receiving ones, they concluded that IL-38 can act as an immunomodulatory agent in the immunopathogenesis of pulmonary fibrosis [184].

Results from a newly conducted study aimed at investigating circulating concentrations of IL-38 in patients with severe acute respiratory syndrome related coronavirus-2 (*SARS-CoV-2*) infection, they reported that there was a negative correlation between serum levels of IL-38 and disease severity [185]. Results from their in vitro study indicated the significant anti-inflammatory effects of IL-38 through inhibiting poly (I:C)-induced overproduction of pro-inflammatory cytokines and amelioration in pulmonary inflammation. In summary, these results indicate the promising therapeutics roles of IL-38 in patients with severe phases of *SARS-CoV-2* infection [185]. Considering the inhibitory effects of IL-38 on CRP in inflamed tissues, it could be a promising molecular therapeutic target for the cases with *SARS-CoV-2* infection [186,187].

##### Lung Adenocarcinoma

As a type of NSCLC, lung adenocarcinoma roughly accounts for a large part (50%) of lung malignancies. Thus, there is an urgent need for early diagnosis and effective immunotherapeutic strategies for lung malignancies. Takada et al. carried out a study to investigate whether or not IL-38 plays a carcinogenic role in the immunopathogenesis of lung adenocarcinoma. They also reported high expression levels of IL-38 in high tumor grades, the presence of pleural and vessel invasions, and advanced stages of NSCLC [188].

Results showed that the prognosis of lung adenocarcinoma has an inverse correlation with the expression levels of IL-38. Accordingly, patients with positive PD-L1 showed higher expression levels of IL-38, resulting in the impairment of acute inflammation by the suppression of CD8^+^ and CD4^+^ T cells [55,58,189]. Thus, overexpression of IL-38 can lead to a poor prognosis for lung adenocarcinoma by showing inhibitory effects on IL-36. Contrary to lung adenocarcinoma, IL-38 is used as therapeutics aimed at attenuating the cytokine release syndrome in inflammatory respiratory disorders [188,189].

Results from another newly conducted study by Fumihiko Kinoshita et al. [190] that aimed at investigating the association of IL-38 and tumor growth in the microenvironment of lung cancer demonstrated a correlation between higher concentrations of IL-38 and poor prognosis in lung adenocarcinoma. In their experimental study, they established IL-38-plasmid-transfected Lewis lung carcinoma cells (LLC-IL-38), and CD8^+^ lymphocyte depletion models to examine possible relationships between the concentrations of IL-38 and CD8^+^ lymphocytes. They reported in vivo increased tumor growth, decreased tumor-infiltrating CD8^+^ lymphocytes, and lower cell proliferation in the LLC-IL-38 model in comparison with the empty vector. Thereafter, they reported that there was an inverse relationship between the expression of IL-38 in cancerous cells and tumor-infiltrating CD8^+^ lymphocytes in lung tumor progression. Therefore, IL-38 can be promisingly recruited as a screening seroimmunobiomarker and a targeted immunotherapeutic approach for lung tumors [190].

##### Chronic Obstructive Pulmonary Disease (COPD) and Acute Lung Injury (ALI)

COPD is a heterogeneous inflammatory disease with different immunopathological pathways (stable state or an acute exacerbated clinical phase). It has been demonstrated that increased levels of IL-38 in the acute states of COPD (to a greater extent) have a negative correlation with plasma levels of CRP and fibrinogen, and a positive correlation with body mass index (BMI) [159,191].

In addition to the diagnostic values of IL-38 in the determination of the severity of COPD [192], the results of a study showed downregulatory impacts of IL-38 on an elastase-induced mouse model of ALI and emphysema, but without changes in the chronic phase [193].

In a study conducted by Matsouka et al., it was revealed that in KO mice (previously sensitized with ovalbumin (OVA)), there was a significant reduction in IL-5 and the number of eosinophils in bronchoalveolar lavage fluid (BALF) [83,194]. Additionally, in another study, they investigated the OVA-induced eosinophilic airway inflammation in C57BL/6N (B6) mice (wild type) and KO mice. They reported higher mRNA expression of IL-38 in lung tissues of OVA challenged wild type mice compared to the saline challenged wild type mice. In addition, there were attenuated levels of eosinophils and IL-5 in BALF samples of KO mice. In summary, it seems that IL-38 could be a promising seroimmunobiomarker to enhance the IL-5 associated OVA-induced eosinophilic inflammation [195]. Correspondingly, in a clinical study, it was reported that there was a negative correlation between the expression levels of IL-38 and the immunopathogenesis of pulmonary embolism. In addition, binary logistic regressions depicted a negative correlation among the risk factors of acute exacerbated COPD. Hence, increased levels of IL-38 could attenuate the expression levels of fibrinogen and CRP, and totally diminish inflammation. Therefore, the results of this study imply the predicting roles of IL-38 in acute exacerbated COPD with pulmonary embolism [196].

These aforementioned results are in accordance with the results from a recently conducted study by DU Wei-Huan et al. [100] that aimed at investigating the correlation of IL-38 concentration and pulmonary function in 107 patients with COPD. Compared to the healthy group, a higher concentration of IL-38 in patients was correlated with lung malfunction. Therefore, these results introduce IL-38 as a screening seroimmunobiomarker for the determination of the severity of the diseases, especially in patients with acute exacerbated COPD [100].

##### Drug-Induced ILD

Lung diseases are classified into a diverse group of respiratory diseases with various and complex clinical manifestations. One of the most prevalent clinical manifestations in patients with ILD is worsened respiratory syndromes accompanied by IPF (progressive-fibrosing phenotype) [197]. As several drugs, some immunological reactions, and inhalation of asbestos or silica dust are involved in the etiopathogenesis of ILD, prognosis and diagnosis of ILD are complicated. Hence, rheumatologists, pulmonologists, immunologists, pathologists, radiologists, internal specialists, and even optometrists should collaborate to provide an accurate diagnosis [198]. Infiltration of myofibroblasts and pneumocytes into the interstitial space of the lung tissues, followed by the secretion of the pro-inflammatory cytokines, can result in the demolition of functional lung tissue and replacement of fibrosis [199].

IL-38 was immunohistochemically investigated in an experimental study after the administration of an anti-human IL-38 monoclonal antibody (clone H127C) on lung tissues of patients with IPF and drug-induced ILD. M. Tominaga et al. reported the overexpression of IL-38 in both case groups in comparison with the control subjects, introducing IL-38 as a screening factor for the deterioration of disease progression (acute/chronic inflammation) as well as a prognostic seroimmunobiomarker for alveolar tissue damage [200,201].

Results of another newly published study conducted by Yu-sen Chai et al. reported elevated levels of IL-38 and TH17 cells of CD4^+^ in serum samples of patients with acute respiratory distress syndrome (ARDS) compared to the control ones. The concentration of IL-38 in serum was positively and significantly correlated with TNF-α, IFN-γ, CXCL-1, CXCL-8, IL-2, IL-4, IL-5, IL-6, IL-17A, and IL-27 [202]. Thereafter, in their cecal ligation puncture (CLP) animal models, there was a remarkable increase in the expression levels of IL-38 in comparison with the control and LPS model mice, concomitant with aggravated ARDS in the IL-38 blockade mouse and a significant decrease in the levels of p-STAT3. In summary, these results showed the therapeutic potential of IL-38 in ARDS [202].

#### 3.2.7. Role of IL-38 in Infectious Diseases

##### Hepatitis

Hepatitis is considered as a life-threatening worldwide health problem. The World Health Organization (WHO) and Centers for Disease Control and Prevention (CDC) present biostatistics about mortality and the morbidity rates of hepatitis, which are deeply rooted in the existence of aberrant genotypes for the hepatitis B virus (HBV) and lack of immunoprophylaxis of infected mothers during their pregnancies [203]. Results of an experimental study reported elevated serum levels of IL-38 in untreated patients with ongoing liver injury. Higher serum levels of IL-38 before treatment indicate a greater probability of virologic response (VR) to telbivudine (LdT) treatment. It was demonstrated that the expression levels of IL-38 diminished to their normal levels after the first 12 weeks of treatment with LdT. Conclusively, IL-38 might be used as a target for monitoring the VR to LdT treatment in patients with hepatitis [204].

In another study, immunobiological factors such as vitamin E, IL-17, IL-37, and IL-38 were assessed in Iraqi male patients with chronic hepatitis B (CHB). Although they reported diminished expression levels of IL-38 and vitamin E in patient groups compared to the healthy controls, they did not recommend monitoring concentrations of IL-38 for the identification of patients with CHB and the assessment of risk stratification [205].

##### Lethal Sepsis

As host systemic inflammatory responses to the invasion of microbial pathogens to the bloodstream, sepsis is considered as one of the major causes of death in hospitalized patients in developing and undeveloped countries mainly due to the indiscriminate usage of antimicrobial drugs and antibiotic resistance. Aberrant inflammatory responses and organ failure after compensatory anti-inflammatory response syndrome (CARS) in septic patients encourage us to investigate more efficacious immunotherapeutic-based solutions [36].

There are increased concentrations of IL-38 in the serum of sepsis patients. Like asthmatic patients, IL-38 decreases the expression levels of inflammatory agents such as IL-6, IL-10, IL-17, IL-27, TNF-α, CXCL1, CCL2, and Treg lymphocytes in septic patients. This signaling reduced organ damages in lung, liver (like concanavalin A-induced liver injury), and kidney in septic patients [36,206]. Elevated serum levels of IL-38 in experimental models and clinical trials of sepsis could alleviate the severity of sepsis and sepsis-related annual mortalities [60]. As such, improved survival rate, reduced tissue damage, and reduced inflammation have been reported in pediatric and adult patients with sepsis [60,207].

Therefore, IL-38 can be considered as a promising therapeutic and diagnostic approach, leading to the decreased invasion of severe pathogens to the bloodstream [60]. In a CLP mouse model-based study, there were increased serum levels of IL-38 in sepsis-induced ALI. Compared to sepsis-induced intrapulmonary ALI, higher serum levels of IL-38 were reported in sepsis-induced extrapulmonary ALI due to the high ratio of CD4^+^ and Treg. In this study, utilization of IL-38 to neutralize antibodies normally led to an exacerbation in the severity of the disease. Hence, these findings certify the usage of IL-38 as a therapeutic target [208].

In polymicrobial sepsis, IL-38 also leads to a controlled bacterial outgrowth and down-regulation of inflammatory responses, leading to decreased mortality. Administration of anti–IL-38 antibody and recombinant IL-38 for patients with polymicrobial sepsis showed poor prognosis and improved survival rate, respectively. Clinically, anti-sepsis targeted therapeutic effects of IL-38 is amplified when combined with IL-5, IL-7, IL-30, and IL-33 [36].

In another newly published study, Yun Ge et al. conducted a murine-based study aimed at investigating the immunobiological roles of IL-38 in immune responses mediated by CD4^+^ CD25^+^ Tregs were involved in the immunopathophisiology of sepsis and they reported remarkably enhanced immunosuppressive activities of the mentioned cells by IL-38 [209].

##### Aspergillosis

Pulmonary Aspergillosis is considered as a life-threatening fungal diseases in immunocompromised patients. It has been clarified that IL-38 plays an important role in the host defense against *Aspergillus fumigatus*. IL-38 acts as an antagonist for IL-36R in Aspergillosis models of infections. The action of this cytokine should be in balanced responses. IL-38 in regulated inflammation shows good results, however, IL-38 in an uncontrolled situation leads to the severity of pathogenesis [34,67].

## 4. Future Directions and Conclusions

Recruitment of the most efficient prognostic, diagnostic, and therapeutic approaches with fewer side effects are of high clinical importance to eradicate immune-mediated diseases, and improve quality of life (QOL) for the patients. Hence, it would be more practically effective to focus on confirmed clinical and diagnostic applications of seroimmunobiomarkers acquired from laboratory-based results. IL-38 is the latest known member of the IL-1 superfamily (IL-36 sub-family) [210,211]. Functionally, IL-38 is a dose (concentration), microenvironment, and receptor-dependent cytokine, acting as pro-inflammatory and anti-inflammatory factors, although the exact role of IL-38 remains unclear. IL-38 binds to the IL-36 receptor and mainly provides anti-inflammatory effects (on PBMCs) similar to IL-36RAs. According to the results acquired from the most updated studies, researchers believe that several natural agents like luteolin (presented in olive) can stimulate the anti-inflammatory activities of IL-38 and can inhibit local inflammation through a reduction in the intracellular levels of reactive oxygen species (ROS) [212].

As reviewed in this study, alterations in serum levels of IL-38 played a major role in the immunopathogenesis of many diseases, in which patients had been previously faced with a wide array of problems due to the lack of prognosis or accurate diagnosis and improper treatment [213,214,215,216,217,218,219,220,221]. At the time of the literature review for this systematic review study, as there is a global pandemic of *Coronavirus*, it has been demonstrated that anti-inflammatory properties of IL-38 can be considered as recently-emerged pharmacotherapies for alleviation of the inflammatory symptoms of patients with *SARS-CoV-2* infection [186,222].

Dealing with all of the above-mentioned frustrating challenges, manipulation of the immune system by triggering several seroimmunobiomarkers such as IL-38 can be rationalized. These immunomodulation-based approaches remind us of the proficiency of immunotherapeutics such as immune gene-based therapy of various pathologic states. In addition, they can open up promising windows to “precision medicine (personalized medicine/individualized medicine)”, especially for immune-mediated disorders with optimal clinical outcomes. Correspondingly, personalized medicine seeks life-long goals that are reached by choosing the “right patients” (investigation of genetic predisposition) and the “right treatment”, which should be administrated at the “right time”. To achieve these potential benefits, the clinical applications of promising seroimmunobiomarkers should be prioritized. Further investigations and collaborations between laboratory scientists, internal medicine specialists, infectious diseases specialists, oncologists, cardiologists, pulmonologists, gynecologists, metabolic disorders specialists, skin specialists, rheumatologists, immunologists, molecular biologists, disease-specific biomarker scientists, gene therapists, basic medical researchers, and health system managers are highly recommended.

## Figures and Tables

**Figure 1 ijms-22-12552-f001:**
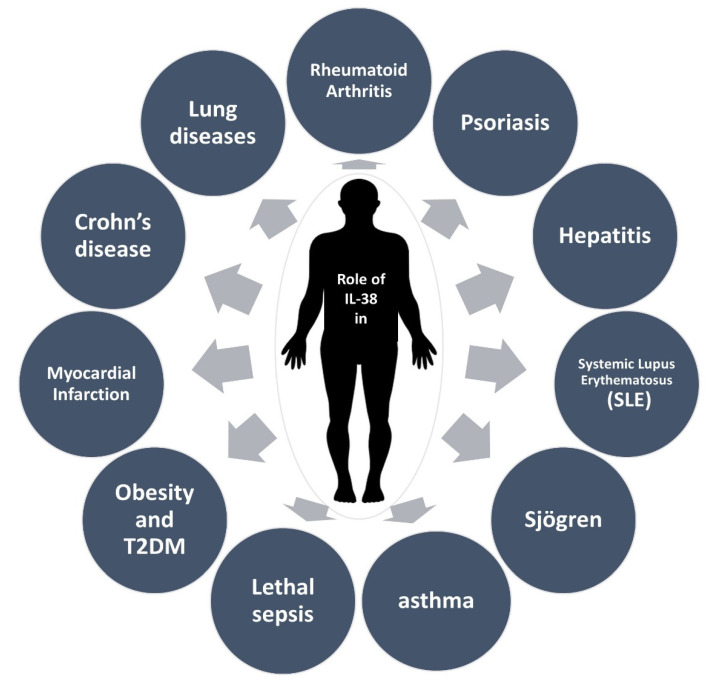
Involvement of IL-38 in immune-mediated diseases. Created by Esmaeilzadeh et al.

**Figure 2 ijms-22-12552-f002:**
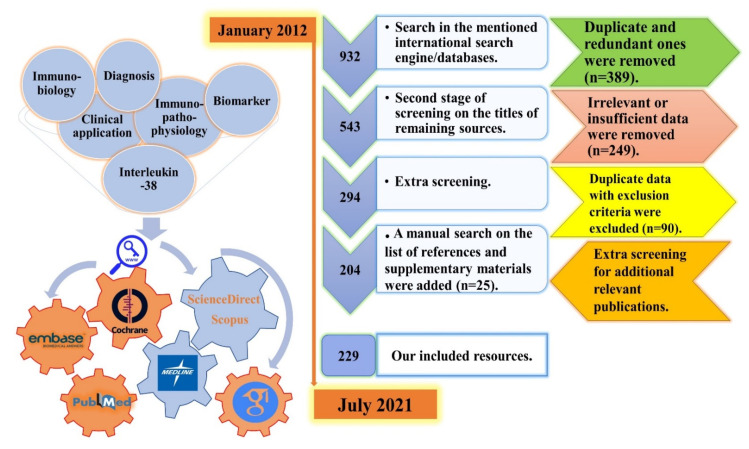
Search strategy according to the PRISMA guideline (PRISMA-P extension 2020 statement). Created by Esmaeilzadeh et al.

**Figure 3 ijms-22-12552-f003:**
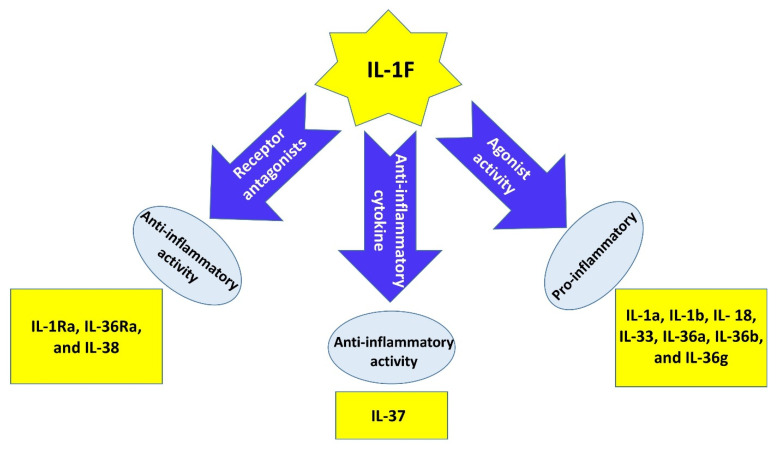
Subtypes of the IL-1 family and their different functions. Created by Esmaeilzadeh et al. The immune functions of members of the IL-1 superfamily depend on their receptor.

**Figure 4 ijms-22-12552-f004:**
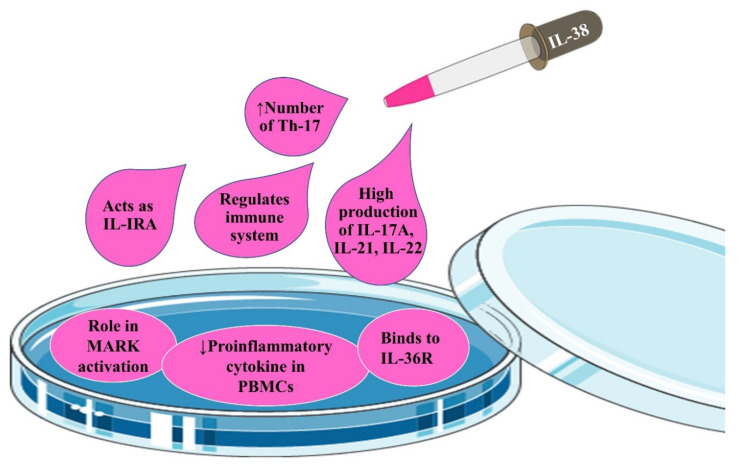
Immunobiological activities of IL-38 in host immune system. Created by Esmaeilzadeh et al. IL-38 acts as an antagonist for the pro-inflammatory properties of IL-36 in PBMCs and as an agonist for inflammatory responses in DCs.

**Figure 5 ijms-22-12552-f005:**
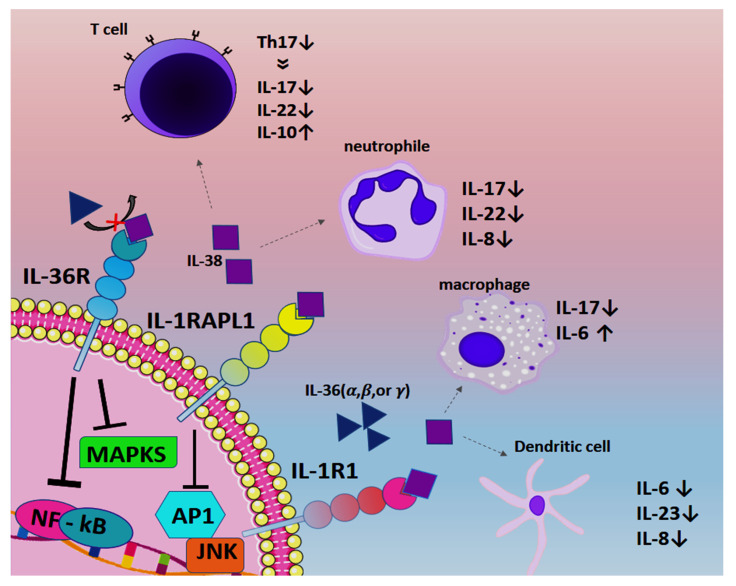
Interactions between IL-38 and involved immunological cells. Created by Esmaeilzadeh et al. Three important receptors of IL-38 and their immunomodulatory effects alter the secretion of pro-inflammatory and anti-inflammatory cytokines. Hence, IL-38 acts as a two-edged sword in immune microenvironments.

**Table 1 ijms-22-12552-t001:** Role of IL-38 in Immune-Mediated Diseases.

Immune-Mediated Diseases	Immunological Roles of IL-38	Reference
Inflammatory Bowel diseases (IBD)	Protective effect in IBD, through production of pro-inflammatory cytokines from macrophages, and a promising immunotherapeutic target in IBD.	[91]
Acne Vulgaris	Exacerbation of skin inflammation.	[92]
Behcet’s Disease (BD)	Exacerbation of eye involvement, and a protective anti-inflammatory role in BS.	[93,94]
Intervertebral Disc Degeneration (IVDD)	Therapeutic roles through alleviation of the inflammatory responses and the degeneration of nucleus pulpous cells via inhibition of the NF-κB signaling pathway.	[95]
Alzheimer	Novel biochemical marker with anti-inflammatory activities.	[96]
Ischemic stroke	Novel early predictor factor for ischemic stroke prognosis.	[97]
Autism spectrum disorder	Therapeutic role through inhibition of activation of human microglia.	[98]
Thyroid-associated ophthalmopathy (TAO)	Protective role in TAO, a promising marker of TAO disease activity, and a potential target for TAO therapy.	[99]
Multiple sclerosis (MS)	Development of through attenuated inflammatory conditions in early stages of MS.	[100]
Experimental autoimmune encephalomyelitis (EAE)	Promotion of inflammation in the central nervous system (CNS).	[101]
Candidiasis	Dose-response reduction in *Candida*-induced T helper 17 responses.	[102]
Arthritis	Significant reduction in clinical inflammation and attenuated severity in mouse models of arthritis.	[103]
Systemic sclerosis	Role in the pathogenesis of systemic sclerosis.	[104]
Atopic dermatitis (eczema)	Prognostication of atopic severity and its inflammatory state in atopic sufferers.	[105]
Osteoporosis	Inhibited proliferation of BMSCs and inhibited apoptosis of osteoblasts by regulating the PI3K/Akt/GSK3β/NFATc1 signaling pathway.	[106]
Asthma	Development of a regulatory cytokine-based treatment for allergic asthma.	[107]
Chronic inflammatory demyelinating polyneuropathy (CIDP)	Making a compensatory mechanism to reduce inflammatory processes in these patients.	[108]
Atopic dermatitis, allergic asthma, and allergic rhinitis	Therapeutic potential in the regulation of allergy asthma, and allergic rhinitis.	[109]
Brucellosis	Progression from acute into the chronic forms of brucellosis.	[110]
inflammatory diseases (psoriasis, rheumatoid arthritis, gout, systemic lupus erythematosus, and Crohn’s disease)	Involved in the pathogenesis of inflammatory diseases.	[111]

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
