# Peer review of "Immunobiological Properties and Clinical Applications of Interleukin-38 for Immune-Mediated Disorders: A Systematic Review Study"

_ijms, 2021, doi:10.3390/ijms222212552_

Round 1
Reviewer 1 Report
The review article by Esmaeilzadeh et al., entitled “Immunobiological Properties and Clinical Applications of Interleukin-38 for Immune-mediated Disorders; A Systematic Review Study” discussed and summarized the immunobiological mechanisms, diagnostic and therapeutic potentials of IL-38.
Overall the manuscript is highly informative. The searching of data and their inclusion/exclusion criteria is extensive and impressive. Critical and systematic reviews like this would surely help researchers in the field to get updated information at one place and the article also falls under the scope of the IJMS.
Though the search criteria are scientific and the article is loaded with information, many sentences don’t make proper sense. In many occasions, some paragraphs feel fragmentary and often logical rhythm is missing. There are numerous grammatical errors which create confusions and in many instances made the sentences almost impossible to understand what the authors actually meant to say. I therefore strongly suggest that the authors have the entire manuscript review/edit by a native/professional English speaking scientist before further submission to any journal. The manuscript has potential, but the language and writing style must be improved. I stopped reading after page 8, as I observed plenty of mistakes that need to be corrected. Therefore, the manuscript can’t be accepted in its current form for publication but can be reconsidered later after edits/corrections.
- Line 34-35: Controversial functions of IL-38, disputably depend on physiologic or pathologic microenvironments, optimal dosage, and involved receptors…the sentence doesn’t make any sense/incomplete
- Line 79: Totally, IL-1 super-….replace Totality with In total
- Line 93-98: In this review, for this purpose, the authors firstly provide information on immunobiol-93 ogy (the structure, receptors, mechanism of signaling, and functional roles) of IL-38. In addition, secondly, they investigate and summarize the role of IL-38 in immunopathophisiology involved in the microenvironment of multiple diseases. Finally, they analyze and discuss the proficiency of IL-38-based therapeutic strategies whether IL-38 cytokine therapy and IL-38 immune gene therapy can be served as a new promising method for the treatment of several diseases….rephase/rewrite the paragraph. Change “they” with “we”.
- Line 99-103: Opening optimistic insights into more collaborations between physicians and laboratory scientists, can end to a successful adjustment of laboratory-based data with clinical outcomes…what the authors tried to mean by this?
- Regardingly, they, bi-functionally can highlight the importance of pre-clinical and basic medical studies in deciphering the secrets of more accurate medical decisions, and optimal clinical outcomes for clinicians. Also, they hopefully increase life expectancy and qualify patients’ lifestyles…rewrite the sentence(s) also clear who are they?
- Line 105-108: This present systematic review study was performed according to the Preferred Report-106 ing Items for Systematic reviews and Meta-analyses (PRISMA) statement guidelines. A global 107 search strategy was conducted for this study based on previously mentioned purposes….provide reference.
- Line 175-176: “According to the structure and type of study (systematic review), there is not any need to be investigated by the ethical committee” Re-write the sentence.
- In the abstract the authors mentioned the search was conducted on articles published during “January 2012 till July 2021” (line 34) but in the line 109-111 (1. Literature Search Strategy) the authors mentioned the timeline as January 2012 till June 2021. Mention the correct timeline.
- Line 183-184: The main cell 183 source of IL-1 such as TNF, is activated mononuclear hs…what is hs?
- Line 200-201: In another classification, the IL-1 family is functionally divided into two groups (based on pro-inflammatory activities and anti-inflammatory activities), which IL-38, IL-1Ra, IL- 36Ra, and IL-37 are in the latter group [54]….rewrite the sentence.
- Line 216-217: As said before, in healthy people, IL-38 is not only expressed normally in the skin, spleen, 216 tonsils, thymus, heart, and liver of the fetus [64, 65], but lower levels of this cytokine in tissues, 217 do not have a specific role in immunity processes, too [64]…re-write the sentence.
- Line 220-221: Additionally, IL-38 is an antagonism of IL-36, explaining the anti-inflammatory effects of IL-38 on immune cells…re-write the sentence.
Author Response
Dear Editor-in-Chief in the International Journal of Molecular Science
Jovial Greetings for the day.
Hope this email finds you well.
First and foremost, on behalf of the authors, I would like to warmly appreciate you, and your specialist reviewers, for your time and consideration on the evaluation of our manuscript.
I believe that our manuscript (manuscript number:) was concisely reviewed by the most knowledgeable and resourceful reviewers in the revision team of your journal.
Below, you see all of the comments and our responses. It is worthy to mention that all of the new changes are done in the content of the manuscript with track changes (in red color in the font without any highlights) as requested in your email. Additionally, all of the revisions were made, but the page number or line numbering in some points were changed after deletion/insertions.
According to the reviewers' comments, their final decisions, and our responses, hope these explanations and responses are enough to meet the criteria for eligibility for possible (final) publication.
I am impatiently looking forward to your last decision on our manuscript.
Sincerely yours.
Dr. Abdolreza Esmaeilzadeh
Corresponding author
Comments and Suggestions for Authors
Reviewer 1
The review article by Esmaeilzadeh et al., entitled “Immunobiological Properties and Clinical Applications of Interleukin-38 for Immune-mediated Disorders; A Systematic Review Study” discussed and summarized the immunobiological mechanisms, diagnostic and therapeutic potentials of IL-38.Overall the manuscript is highly informative. The searching of data and their inclusion/exclusion criteria is extensive and impressive. Critical and systematic reviews like this would surely help researchers in the field to get updated information at one place and the article also falls under the scope of the IJMS.
Thanks for the comment and your precise evaluation. Definitely, we did our best to include and explain the main contribution of this article to the future studies. In this round of revisions, we reviewed our search strategy and did our final searches to make our manuscript more updated.
Though the search criteria are scientific and the article is loaded with information, many sentences don’t make proper sense. In many occasions, some paragraphs feel fragmentary and often logical rhythm is missing. There are numerous grammatical errors which create confusions and in many instances made the sentences almost impossible to understand what the authors actually meant to say. I therefore strongly suggest that the authors have the entire manuscript review/edit by a native/professional English speaking scientist before further submission to any journal. The manuscript has potential, but the language and writing style must be improved. I stopped reading after page 8, as I observed plenty of mistakes that need to be corrected. Therefore, the manuscript can’t be accepted in its current form for publication but can be reconsidered later after edits/corrections.
Thanks for the comment. We highly respect your concern. In this round of revision, we evaluated all of the manuscript and tried to increase cohesion and coherency of the sentences. All of the English improvement and grammatical errors were corrected by a native English-speaking editor throughout the manuscript. Additionally, we used one software for grammatical corrections and improvement in writing style to make the language of our manuscript closer to the real science.
- Line 34-35: Controversial functions of IL-38, disputably depend on physiologic or pathologic microenvironments, optimal dosage, and involved receptors…the sentence doesn’t make any sense/incomplete
Thanks for the comment. Mentioned sentence was completely re-written as you requested.
- Line 79: Totally, IL-1 super-….replace Totality with In total
Thanks for the comment. Replacement is done in the mentioned page and line number as you requested.
- Line 93-98: In this review, for this purpose, the authors firstly provide information on immunobiology (the structure, receptors, mechanism of signaling, and functional roles) of IL-38. In addition, secondly, they investigate and summarize the role of IL-38 in immunopathophisiology involved in the microenvironment of multiple diseases. Finally, they analyze and discuss the proficiency of IL-38-based therapeutic strategies whether IL-38 cytokine therapy and IL-38 immune gene therapy can be served as a new promising method for the treatment of several diseases….rephrase/rewrite the paragraph. Change “they” with “we”.
Thanks for the comment. Aim of study at the end of introduction part was completely re-phrased and re-written as you requested. We tried to make the sentences shorter and reduce the ambiguities. Replacement is done in the mentioned page and line number.
- Line 99-103: Opening optimistic insights into more collaborations between physicians and laboratory scientists, can end to a successful adjustment of laboratory-based data with clinical outcomes…what the authors tried to mean by this?
Thanks for the comment. We definitely value your concerns on misunderstanding. We, on the one hand, tried our best to shorten long sentences and make them more simplified and remove ambiguities in the last sentences of introduction. On the other hand, we tried to show the wide range of readership to this manuscript that how they can benefit from data. Aim of study at the end of introduction was completely re-phrased and re-written as requested.
- Regardingly, they, bi-functionally can highlight the importance of pre-clinical and basic medical studies in deciphering the secrets of more accurate medical decisions, and optimal clinical outcomes for clinicians. Also, they hopefully increase life expectancy and qualify patients’ lifestyles…rewrite the sentence(s) also clear who are they?
Thanks for the comment. We definitely value your concerns on misunderstanding. We, on the one hand, tried our best to shorten long sentences and make them more simplified and remove ambiguities in the last sentences of introduction. On the other hand, we wanted to make promising recommendations to broaden knowledge among peers in medicine and basic medical sciences. Aim of study at the end of introduction was completely re-phrased and re-written as requested.
- Line 105-108: This present systematic review study was performed according to the Preferred Reporting Items for Systematic reviews and Meta-analyses (PRISMA) statement guidelines. A global search strategy was conducted for this study based on previously mentioned purposes….provide reference.
Thank you for updated views. The reference was mentioned in the “Methodology” part.
- Line 175-176: “According to the structure and type of study (systematic review), there is not any need to be investigated by the ethical committee” Re-write the sentence.
Many thanks for the comment. The sentence was re-written in aforementioned section.
- In the abstract the authors mentioned the search was conducted on articles published during “January 2012 till July 2021” (line 34) but in the line 109-111 (1. Literature Search Strategy) the authors mentioned the timeline as January 2012 till June 2021. Mention the correct timeline.
Many thanks for your precise look. It was an erroneous report. And time interval in the timeline was corrected as July 2021. This point is checked wherever it is mentioned.
- Line 183-184: The main cell 183 source of IL-1 such as TNF, is activated mononuclear hs…what is hs?
Many thanks for your comment. It was a typeset error and cleared. The final sentence was also re-written.
- Line 200-201: In another classification, the IL-1 family is functionally divided into two groups (based on pro-inflammatory activities and anti-inflammatory activities), which IL-38, IL-1Ra, IL- 36Ra, and IL-37 are in the latter group [54]….rewrite the sentence.
Thanks for the comment. This paragraph was completely re-phrased and re-written as you requested. We tried to make the sentences shorter and reduce the ambiguities with a clear presentations of various classifications. Replacement is done in the mentioned page and line number.
- Line 216-217: As said before, in healthy people, IL-38 is not only expressed normally in the skin, spleen, 216 tonsils, thymus, heart, and liver of the fetus [64, 65], but lower levels of this cytokine in tissues, 217 do not have a specific role in immunity processes, too [64]…re-write the sentence.
Thanks for the comment. This paragraph was completely re-phrased and re-written with proper usage of connections and grammatical points as you requested.
- Line 220-221: Additionally, IL-38 is an antagonism of IL-36, explaining the anti-inflammatory effects of IL-38 on immune cells…re-write the sentence.
Thanks for the comment. This paragraph was completely re-phrased and re-written with proper usage of connections and grammatical points as you requested.
Submission Date
16 August 2021
Date of this review
27 Aug 2021 08:06:01
Reviewer 2 Report
This review try to summarize the actual knowledge of IL-38.
Critique:
The article is too long. Its writing is messy and confusing and makes it difficult to read. Major changes has to be done in order to reduce the dimension of the article and to improve the understanding of it. Lenguage must be brief and concise. Some examples are showed below:
Line 53 to 57 - "The chronicity of the mentioned disorders has still remained a problematic challenge for the health system. These points, transparently indicate that patients with immune-mediated disorders are sensibly coping with poor lifestyles, and suffering from complex clinical manifestations in their life-expectancy due to poor diagnosis or prognosis.
The chronicity of these type of disorders has still remained a problematic challenge for the health system. In addition, patients with immune-mediated disorders experience poor lifestyles and suffer for complex clinical manifestations in their life-expentancy due to poor diagnosis and prognosis.
Line 78 to 81 - "Interleukin-1 (IL-1) family, previously named human leukocytic pyrogen, are considered as key players in interactions of both innate and adaptive immune systems. Totally, IL-1 super-family encircles 4 subgroups and 11 members. These subgroups enclose IL-1, IL-18, IL-33, and IL-36 subfamilies [8-10]."
Interleukin-1 (IL-1) family are essential players in the interactions of both innate and adaptative immune systems. IL-1 super-family encircles 4 main subgroups (IL-1, IL-18, IL-33 and IL-36 subfamilies) with a total of 11 members, all of them playing a key role in the initiation and regulation of the immune response in immune-mediated diseases.
Line 94 to 98 - "In addition, secondly, they investigate and summarize the role of IL-38 in immunopathophisiology involved in the microenvironment of multiple diseases. Finally, they analyze and discuss the proficiency of IL-38-based therapeutic strategies whether IL-38 cytokine therapy and IL-38 immune gene therapy can be served as a new promising method for the treatment of several diseases."
Secondly, they investigate and summarize the role of IL-38 in the immunopathophisiology of multiple diseases. Finally, they analyze and discuss the proficiency of IL-38-based therapeutic strategies, such us IL-38 cytokine therapy and IL-38 immune gene therapy, as new promising methods for the treatment of several diseases.
Line 105 to 159 - Multiple changes should be done in the methodology section:
- Replace "Search Method" for "Methodology".
- As a suggestion, merge subheadings 2.1 and 2.3.
- Subheading 2.2 is difficult to understand. Authors should consider rewriting it completely.
- Figure 1 should be mentioned after "...statement guidelines" (line 107).
- Line 115. Replace "that" for "which" and "in the next part" for "bellow" or "in the next subsection".
- Line 149. Replace the first "stage" for "step".
Subsection 3.1. - This subheading is too long. Results should be focused on the IL-38 molecule. Irrelevant information regarding the IL-1 family should be removed. As a suggestion "The IL-1 family consists of several molecules that are categorized into different groups based on their effects on receptors. Some of these molecules are generally known as agonists (IL-1α, IL-1β, IL- 18, IL-33, IL-36α, IL-36β, and IL-36γ), whereas other ones are considered as Receptor Antagonists (RA). IL-1RA, IL-36RA (IL-1R6A), and IL-38 are prime examples of RA and one member is anti-inflammatory (IL-37) (fig.2) [32-53]." could be the first paragraph with minnor changes:
"The IL-1 family consists of several molecules that are categorized into different groups based on their effects on receptors. In this sense, some of these molecules are generally known as agonists (IL-1α, IL-1β, IL- 18, IL-33, IL-36α, IL-36β, and IL-36γ), whereas other ones are considered as antagonists (IL-1RA, IL-1R6A, and IL-38). Another member, IL-37, is anti-inflammatory (fig.2) [32-53]."
Subsection 3.1.1 - This section could be reduced to 2- 5 lines.
Line 243 - Replace "3.1.1." for "3.1.2.".
Line 296 - Replace "3.1.2." for "3.1.3.".
Subsection 3.2.1. - Figure 5 should be replace by a summary table with 3 columns: Disease, Role of IL-38 and References.
Although the field of study is completely different, the reviewer suggests a recent review published by Inés Robles Mendo et al. (PMID: 34410512) as a good model to follow. This review presents a brief and concise language, a good structure and a good number of tables and figures for its follow-up and understanding.
Author Response
Dear Editor-in-Chief in the International Journal of Molecular Science
Jovial Greetings for the day.
Hope this email finds you well.
First and foremost, on behalf of the authors, I would like to warmly appreciate you, and your specialist reviewers, for your time and consideration on the evaluation of our manuscript.
I believe that our manuscript (manuscript number:) was concisely reviewed by the most knowledgeable and resourceful reviewers in the revision team of your journal.
Below, you see all of the comments and our responses. It is worthy to mention that all of the new changes are done in the content of the manuscript with track changes (in red color in the font without any highlights) as requested in your email. Additionally, all of the revisions were made, but the page number or line numbering in some points were changed after deletion/insertions.
According to the reviewers' comments, their final decisions, and our responses, hope these explanations and responses are enough to meet the criteria for eligibility for possible (final) publication.
I am impatiently looking forward to your last decision on our manuscript.
Sincerely yours.
Dr. Abdolreza Esmaeilzadeh
Corresponding author
Comments and Suggestions for Authors
Reviewer 2
This review try to summarize the actual knowledge of IL-38.
Critique:
The article is too long. Its writing is messy and confusing and makes it difficult to read. Major changes has to be done in order to reduce the dimension of the article and to improve the understanding of it. Language must be brief and concise.
Thanks for the comment. We highly respect your concern. In case of the length of manuscript, around 1800 words is reduced after making all revisions. In this round of revision, we evaluated all of the manuscript and tried to increase cohesion and coherency of the sentences. All of the English improvement and grammatical errors were corrected by a native English-speaking editor throughout the manuscript. Additionally, we used one software for grammatical corrections and improvement in writing style to make the language of our manuscript closer to the real science.
Some examples are showed below:
Line 53 to 57 - "The chronicity of the mentioned disorders has still remained a problematic challenge for the health system. These points, transparently indicate that patients with immune-mediated disorders are sensibly coping with poor lifestyles, and suffering from complex clinical manifestations in their life-expectancy due to poor diagnosis or prognosis.
The chronicity of these type of disorders has still remained a problematic challenge for the health system. In addition, patients with immune-mediated disorders experience poor lifestyles and suffer for complex clinical manifestations in their life-expectancy due to poor diagnosis and prognosis.
Thanks for the comment. This paragraph was completely re-phrased and re-written as you requested. Replacement is done in the mentioned page and line number.
Line 78 to 81 - "Interleukin-1 (IL-1) family, previously named human leukocytic pyrogen, are considered as key players in interactions of both innate and adaptive immune systems. Totally, IL-1 super-family encircles 4 subgroups and 11 members. These subgroups enclose IL-1, IL-18, IL-33, and IL-36 subfamilies [8-10]."
Interleukin-1 (IL-1) family are essential players in the interactions of both innate and adaptative immune systems. IL-1 super-family encircles 4 main subgroups (IL-1, IL-18, IL-33 and IL-36 subfamilies) with a total of 11 members, all of them playing a key role in the initiation and regulation of the immune response in immune-mediated diseases.
Thanks for the comment. This paragraph was completely re-phrased and re-written as you requested. Replacement is done in the mentioned page and line number.
Line 94 to 98 - "In addition, secondly, they investigate and summarize the role of IL-38 in immunopathophisiology involved in the microenvironment of multiple diseases. Finally, they analyze and discuss the proficiency of IL-38-based therapeutic strategies whether IL-38 cytokine therapy and IL-38 immune gene therapy can be served as a new promising method for the treatment of several diseases."
Secondly, they investigate and summarize the role of IL-38 in the immunopathophisiology of multiple diseases. Finally, they analyze and discuss the proficiency of IL-38-based therapeutic strategies, such us IL-38 cytokine therapy and IL-38 immune gene therapy, as new promising methods for the treatment of several diseases.
Thanks for the comment. This paragraph was completely re-phrased and re-written as you requested. Replacement is done in the mentioned page and line number.
Line 105 to 159 - Multiple changes should be done in the methodology section:
- Replace "Search Method" for "Methodology".
Thanks for the comment. Replacement is done in the mentioned page and line number.
- As a suggestion, merge subheadings 2.1 and 2.3.
Thanks for the comment. Replacement and merging are done in the mentioned page and line number with reduced word counts and proper placement of the figure.
- Subheading 2.2 is difficult to understand. Authors should consider rewriting it completely.
Thanks for the comment. This paragraph was completely re-phrased and re-written as you requested. It is tried to make “inclusion” and “exclusion” criteria more comprehensive and understandable with reduced word counts and proper usage of grammatical points. Replacement is done in the mentioned page and line number.
- Figure 1 should be mentioned after "...statement guidelines" (line 107).
Thanks for the comment. Replacement is done in the mentioned page and line number.
- Line 115. Replace "that" for "which" and "in the next part" for "bellow" or "in the next subsection".
Thanks for the comment. Replacement is done in the mentioned page and line number.
- Line 149. Replace the first "stage" for "step".
Thanks for the comment. Replacement is done in the mentioned page and line number.
Subsection 3.1. - This subheading is too long. Results should be focused on the IL-38 molecule. Irrelevant information regarding the IL-1 family should be removed. As a suggestion "The IL-1 family consists of several molecules that are categorized into different groups based on their effects on receptors. Some of these molecules are generally known as agonists (IL-1α, IL-1β, IL- 18, IL-33, IL-36α, IL-36β, and IL-36γ), whereas other ones are considered as Receptor Antagonists (RA). IL-1RA, IL-36RA (IL-1R6A), and IL-38 are prime examples of RA and one member is anti-inflammatory (IL-37) (fig.2) [32-53]." could be the first paragraph with minor changes:
"The IL-1 family consists of several molecules that are categorized into different groups based on their effects on receptors. In this sense, some of these molecules are generally known as agonists (IL-1α, IL-1β, IL- 18, IL-33, IL-36α, IL-36β, and IL-36γ), whereas other ones are considered as antagonists (IL-1RA, IL-1R6A, and IL-38). Another member, IL-37, is anti-inflammatory (fig.2) [32-53]."
Thanks for the comment. This paragraph was completely re-phrased and re-written as you requested. Replacement is done in the mentioned page and line number.
Subsection 3.1.1 - This section could be reduced to 2- 5 lines.
Thanks for the comment. I would like to warmly appreciate you for the precise evaluation of our manuscript. We highly respect your comment and understand your concerns about the qualified presentation of data within titles and subtitles. Definitely, we tried our best to make a comprehensive and summarized content in this section. Also, we made a checking throughout the manuscript and synchronize the content. All in all, we hope that this newly edited version is of enough quality and quantity to meet the criteria for further publication.
Line 243 - Replace "3.1.1." for "3.1.2.".
Thanks for the comment. All of the headings and subheadings were checked throughout the manuscript after final insertion/deletion/replacement. Replacement is done in the mentioned page and line number.
Line 296 - Replace "3.1.2." for "3.1.3.".
Thanks for the comment. All of the headings and subheadings were checked throughout the manuscript after final insertion/deletion/replacement. Replacement is done in the mentioned page and line number.
Subsection 3.2.1. - Figure 5 should be replaced by a summary table with 3 columns: Disease, Role of IL-38 and References.
Thanks for the comment. Replacement of aforesaid figure by a summarized table was a very good idea that we eagerly did it. So, now, our manuscript is visually qualified. Afterward, we tried not to omit the figure 5 and not reduce the visualization. We moved it toward introduction as figure 1 due to the proper content that supports it. Additionally, we shortened the sentences in order to make our figure more informative.
Although the field of study is completely different, the reviewer suggests a recent review published by Inés Robles Mendo et al. (PMID: 34410512) as a good model to follow. This review presents a brief and concise language, a good structure and a good number of tables and figures for its follow-up and understanding.
Thanks for your precise and updated views which made our manuscript more upgraded. I would like to sincerely thank you for your comprehensive and interesting views on the topic, which made us to have more creative and innovative minds in our idea and the goal of this study. Honestly-speaking, I enjoyed your scientific and academic views in your comments. It’s a privilege to us to hear such a promising comments from you. Additionally, I would like to make you rest assured that all of the comments were responded, and a second native English-speaking editor kindly evaluated the manuscript and revised all of the necessitated corrections (grammatical and punctuational) throughout the manuscript.
Submission Date
16 August 2021
Date of this review
30 Aug 2021 10:24:10
Reviewer 3 Report
The genesis of this review is the proposal of the Authors to present an overview concerning the immuno-biological molecular mechanisms and therapeutic potentials of IL-38, one of the members of IL-1 family.
For this purpose, the Authors:
- describe the immunobiology of IL-36,
- analyze and discuss the role of IL-38 in the microenvironment of many severe diseases
- analyze and discuss the proficiency of IL-38-based therapeutic strategies whether IL-38 cytokine therapy and IL-38 immune gene therapy can be served as a new promising method for the treatment of such diseases.
This is a very good review. It describes a beautiful synthesis of the so far available data regarding the newly discovered cytokine IL-38. The paper is scientifically accurate, and the conclusions are well supported by the references. Indeed, in my view, the main strength of this review is the exhaustiveness and the quality of the references. Importantly, the numerous citations mentioned provide both specialist and non-specialist readers a well-detailed and clear vision of the topic, in particular concerning the 1. the immuno-biological role of IL-38 and 2. the role played by IL-38 in several inflammatory diseases. Therefore, two of the three main points that the Authors suggest as the core pillars of the manuscript, are well discussed and structured.
On the other hand, the third point, that would allow this review to stand out from other similar works (i.e. "Garraud et al., Cytokine and Growth Factor, 2018" - "Xu & Huang, Front. Immunol.,2018" – “Xie et al., Biomolecules, 2019) represents the main weakness of the paper. Indeed, I believe that this aspect is poorly discussed and developed in the manuscript and, furthermore, in this context the paper is not sufficiently innovative. I find the paper lacks the ability to propose new ideas/hypothesis in terms of the development of 1. research projects and 2. therapies concerning IL-38 as the molecular target of the treatment. I was wondering if the Authors could develop a bit more this point.
Another concern is in my opinion represented by the figures. The manuscript includes five figures, some of which are poorly representative, in particular Fig. 2 (I consider it superfluous) and Fig. 3, which should definitely be improved from a graphical point of view. In addition, captions are very brief, and they do not sufficiently discuss the relative figures, and a figure that presents what the Authors aim to discuss in the third point could be added to the manuscript. Furthermore, because of the huge number of references, I suggest the Authors to add a table where to summarize the main discoveries/observations of the papers they have investigated, to facilitate the reader to untangle the text.
Author Response
Dear Editor-in-Chief in the International Journal of Molecular Science
Jovial Greetings for the day.
Hope this email finds you well.
First and foremost, on behalf of the authors, I would like to warmly appreciate you, and your specialist reviewers, for your time and consideration on the evaluation of our manuscript.
I believe that our manuscript (manuscript number:) was concisely reviewed by the most knowledgeable and resourceful reviewers in the revision team of your journal.
Below, you see all of the comments and our responses. It is worthy to mention that all of the new changes are done in the content of the manuscript with track changes (in red color in the font without any highlights) as requested in your email. Additionally, all of the revisions were made, but the page number or line numbering in some points were changed after deletion/insertions.
According to the reviewers' comments, their final decisions, and our responses, hope these explanations and responses are enough to meet the criteria for eligibility for possible (final) publication.
I am impatiently looking forward to your last decision on our manuscript.
Sincerely yours.
Dr. Abdolreza Esmaeilzadeh
Corresponding author
Comments and Suggestions for Authors
Reviewer 3
The genesis of this review is the proposal of the Authors to present an overview concerning the immuno-biological molecular mechanisms and therapeutic potentials of IL-38, one of the members of IL-1 family.
For this purpose, the Authors:
- describe the immunobiology of IL-36,
- analyze and discuss the role of IL-38 in the microenvironment of many severe diseases
- analyze and discuss the proficiency of IL-38-based therapeutic strategies whether IL-38 cytokine therapy and IL-38 immune gene therapy can be served as a new promising method for the treatment of such diseases.
This is a very good review. It describes a beautiful synthesis of the so far available data regarding the newly discovered cytokine IL-38. The paper is scientifically accurate, and the conclusions are well supported by the references.
So many thanks for your nice and promising words. It is our privilege to have these words from you. We rechecked our manuscript and add the latest references to the first part of “future directions” before “conclusion” aimed at making it more comprehensive and updated. Additionally, we made a checking throughout the manuscript for improvement of all sections.
Indeed, in my view, the main strength of this review is the exhaustiveness and the quality of the references. Importantly, the numerous citations mentioned provide both specialist and non-specialist readers a well-detailed and clear vision of the topic, in particular concerning the 1. the immuno-biological role of IL-38 and 2. the role played by IL-38 in several inflammatory diseases. Therefore, two of the three main points that the Authors suggest as the core pillars of the manuscript, are well discussed and structured.
Many thanks for your comprehensive views on the content, which made us to write our analysis more clarified with a great motivation for more continuity. As it is clear from your words, we deeply comprehend that you are a resourceful reviewer in this field which is our privilege to have these kind of words from you.
On the other hand, the third point, that would allow this review to stand out from other similar works (i.e. "Garraud et al., Cytokine and Growth Factor, 2018" - "Xu & Huang, Front. Immunol.,2018" – “Xie et al., Biomolecules, 2019) represents the main weakness of the paper. Indeed, I believe that this aspect is poorly discussed and developed in the manuscript and, furthermore, in this context the paper is not sufficiently innovative. I find the paper lacks the ability to propose new ideas/hypothesis in terms of the development of 1. research projects and 2. therapies concerning IL-38 as the molecular target of the treatment. I was wondering if the Authors could develop a bit more this point.
We highly respect it. In this revision, we tried to make a comprehensive search and add sentences to each newly added or previously added clinical or experimental study to highlight therapeutic, prognostic, and diagnostic potentials of IL-38. We include all of existent researches.
Another concern is in my opinion represented by the figures. The manuscript includes five figures, some of which are poorly representative, in particular Fig. 2 (I consider it superfluous) and Fig. 3, which should definitely be improved from a graphical point of view. In addition, captions are very brief, and they do not sufficiently discuss the relative figures, and a figure that presents what the Authors aim to discuss in the third point could be added to the manuscript. Furthermore, because of the huge number of references, I suggest the Authors to add a table where to summarize the main discoveries/observations of the papers they have investigated, to facilitate the reader to untangle the text.
We do definitely agree with you. And tried to diminish the word count of our study. In case of number of references, as we declared that this study is the most comprehensive ones from 2012 until 2021, we tried not to skip any related study. In case of figures, they are all designed by PowerPoint software and the authors. None of them are downloaded. Captions have been corrected and to show more detailed contents. Around 1800 words is reduced after making all revisions.
Submission Date
16 August 2021
Date of this review
01 Sep 2021 10:14:00
Round 2
Reviewer 1 Report
Dear authors, even after the review I observed innumerable writing mistakes and errors throughout the manuscript. Several sentences have been deleted instead of correction. I'm afraid if these mistakes persists, the future readers of this article may either won't be able to understand properly or may actually misunderstand several parts of the manuscript. Therefore I again strongly suggest to get the manuscript edited by English speaking scientist (who understand the subject matter) before further submission.
Below are a couple of such examples I observed in first couple of pages:
Presumably, these problems lead to necessitating unraveled notions, and accentuating more efficient researches to settle down mentioned problems...unclear and confusing
Totally, these points lead scientists toward a comprehensive understanding of cellular and molecular mechanisms involved in human immune system at the laboratory benches to reach the most effective therapeutics at the bedside.
The affinity of this cytokine on this receptor is less than two other receptors that are explained as follows
It was observed that bonded IL-38 to receptor on the macrophage surface, inhibited the secretion of IL-6.
Several studies introduce “concentration” as determinant for intensity of immunomodulatory effects of IL-38
Author Response
Dear Editor-in-Chief in the International Journal of Molecular Science
Jovial Greetings for the day.
Hope this email finds you well, and you have gotten vaccinated.
First and foremost, on behalf of the authors, I would like to warmly appreciate you again, and your specialist reviewers, for your time and consideration on the second evaluation of our manuscript.
I believe that our manuscript (manuscript number: ijms-1359758) was concisely reviewed by the most knowledgeable and resourceful reviewers in the revision team of your journal.
Below, you see all of the comments and our responses. It is worthy to mention that all of the new changes are done in the content of the manuscript in blue color in the font (without any highlights) in order to be distinct from first round of revision. Additionally, all of the revisions were made, but the page number or line numbering in some points were normally changed after deletion/insertions.
According to the reviewers' comments, their final decisions, and our responses, hope these explanations and responses are enough to meet the criteria for eligibility for possible (final) publication.
I am impatiently looking forward to your last decision on our manuscript.
Sincerely yours.
Dr. Abdolreza Esmaeilzadeh
Corresponding author
Comments and Suggestions for Authors
Reviewer 1
Dear authors, even after the review I observed innumerable writing mistakes and errors throughout the manuscript. Several sentences have been deleted instead of correction. I'm afraid if these mistakes persists, the future readers of this article may either won't be able to understand properly or may actually misunderstand several parts of the manuscript. Therefore I again strongly suggest to get the manuscript edited by English speaking scientist (who understand the subject matter) before further submission.
Thanks for the comment. We highly respect your concern. In the second round of revision, we re-evaluated all of the manuscript in case of cohesion and coherency of the content. All of the English improvement and grammatical errors were corrected by a native English-speaking scientist (immunologist) throughout the manuscript. Additionally, we used another software for grammatical corrections and improvement in writing style to make the language of our manuscript closer to the real science. In addition, we originally did not omit any sentences or contents. Maybe you have reviewed the edited version by track change. As you and other two reviewers highly recommended us to shorten word counts, we only had tried to lessen the words and in the same time, keep the content with the most proper style of writing. All in all, we would like to make you rest assured that we re-evaluated the manuscript for the second time which you see in blue font (without any highlights).
Below are a couple of such examples I observed in first couple of pages:
Presumably, these problems lead to necessitating unraveled notions, and accentuating more efficient researches to settle down mentioned problems...unclear and confusing.
Thanks for the comment. Mentioned sentence was completely re-written (in blue font without any highlights) as you requested.
Totally, these points lead scientists toward a comprehensive understanding of cellular and molecular mechanisms involved in human immune system at the laboratory benches to reach the most effective therapeutics at the bedside.
Thanks for the comment. Mentioned sentence was completely re-written (in blue font without any highlights) as you requested.
The affinity of this cytokine on this receptor is less than two other receptors that are explained as follows.
Thanks for the comment. Mentioned sentence was completely re-written (in blue font without any highlights) as you requested.
It was observed that bonded IL-38 to receptor on the macrophage surface, inhibited the secretion of IL-6.
Thanks for the comment. Mentioned sentence was completely re-written (in blue font without any highlights) as you requested.
Several studies introduce “concentration” as determinant for intensity of immunomodulatory effects of IL-38.
Thanks for the comment. Mentioned sentence was completely re-written (in blue font without any highlights) as you requested.
Submission Date
16 August 2021
Date of this review
15 Sep 2021 12:18:44
Reviewer 2 Report
Line 98-99 . Replace the sentence by: Recently, different studies have demonstrated multiple roles of IL-38 in several immune-mediated conditions.
Line 152. Replace "ail" by "aim".
Author Response
Dear Editor-in-Chief in the International Journal of Molecular Science
Jovial Greetings for the day.
Hope this email finds you well, and you have gotten vaccinated.
First and foremost, on behalf of the authors, I would like to warmly appreciate you again, and your specialist reviewers, for your time and consideration on the second evaluation of our manuscript.
I believe that our manuscript (manuscript number: ijms-1359758) was concisely reviewed by the most knowledgeable and resourceful reviewers in the revision team of your journal.
Below, you see all of the comments and our responses. It is worthy to mention that all of the new changes are done in the content of the manuscript in blue color in the font (without any highlights) in order to be distinct from first round of revision. Additionally, all of the revisions were made, but the page number or line numbering in some points were normally changed after deletion/insertions.
According to the reviewers' comments, their final decisions, and our responses, hope these explanations and responses are enough to meet the criteria for eligibility for possible (final) publication.
I am impatiently looking forward to your last decision on our manuscript.
Sincerely yours.
Dr. Abdolreza Esmaeilzadeh
Corresponding author
Comments and Suggestions for Authors
Reviewer 2
Line 98-99 . Replace the sentence by: Recently, different studies have demonstrated multiple roles of IL-38 in several immune-mediated conditions.
Thanks for the comment. Mentioned sentence was completely re-written (in blue font without any highlights) as you requested.
Line 152. Replace "ail" by "aim".
Thanks for the comment. It was a typeset error. Mentioned word was completely replaced (in blue font without any highlights) as you requested.
Submission Date
16 August 2021
Date of this review
10 Sep 2021 08:43:34

Round 3
Reviewer 1 Report
Dear authors, I’m sorry to say that I have again read until page 8 (out of 32 pages) and found multiple grammatical and scientific errors. The writing is still of poor quality and I personally can’t recommend publication of this article in its present form. The entire manuscript requires extensive editing to make the scientific contents clear and understandable. Thank you for your understanding.
Line 78-79: These types of studies contribute us in order to reach the most effective therapeutics…the sentence needs rephrasing…you can replace “contribute” with “help”
Line 84-87: Regardingly, molecular immunobiology may provide a clinical basis for further investigations of the therapeutic applications of cytokines [6, 7]…a subject name can’t provide anything…studies on that field can..
Line 92-93: Interleukin-1 (IL-1) superfamily are essential players in the interactions of both innate and adaptive immune systems…again Interleukin-1 (IL-1) superfamily can’t be essential…members of the Interleukin-1 (IL-1) superfamily (e.g., different cytokines) can be..
Line 102-107: Thus, the administration of novel IL-38 gene-based immunotherapeutics can be addressed to support this evidence understanding the complete mechanisms of IL-38 function and it’s effects on different diseases [19-22]…complicated and confusing sentence
Line 111-112: In this review, we firstly provide information on immunobiology (the structure, receptors, mechanism of signaling, and functional roles) of IL-38… the sentence needs rephrasing
Line 120-121: We believe that results of such studies can open optimistic insights into more collaborations between physicians and laboratory scientists… the sentence need rephrasing
Line 124-127: Regardingly, we believe that results of such studies bi-functionally can highlight the importance of pre-clinical and basic medical studies in more accurate medical decisions, and optimal clinical outcomes for clinicians…I don’t understand the meaning of this
Line 127-129: Also, clinical administration or therapeutic usage of seroimmunobiomarkers with reduced side effects can hopefully increase life- expectancy and qualify patients’ lifestyles…I can understand the usage, but what do you mean by administration of seroimmunobiomarkers? Also, what is “qualify patients’ lifestyles”
Line 162: what do you mean by “were totally included”?
Line 167: Studies that investigated secretion of IL-38 after usage of immunostimulants like vaccine, excluded…briefly mention why did you exclude these study reports.
Line 221-222: IL-38 from apoptotic cells is not properly recognized…discuss a bit more on this.
Line 253: IL-38 is present in all types of vertebrates with evolved immune responses [38]…”with evolved immune responses”..I’m not sure if this term is scientifically correct!
Line 254-256: Despite numerous approaches for purification and expression of IL-38 in bacterial systems, Yuan et al successfully reached their hypothesis by using a prokaryotic expression vector (pET-44) and a one-step Ni2+-agarose affinity chromatography in a relatively large quantity of a C-terminus tagged IL-38 [27]…confusing and incorrect statement.
Line 263: Additionally, IL-38 is numerated as an antagonism of for IL-36…what does it mean?
Line 264-266: The anti-inflammatory role of IL-38 is postulated according to the sharing homology with IL-1Ra (an IL-1R1A), and with IL-36RA, making IL-38 as an antagonists with IL-1R1, and IL-36, respectively (20)…confusing sentence.
Author Response
Dear Editor-in-Chief in the International Journal of Molecular Science
Jovial Greetings for the day.
Hope this email finds you well.
First and foremost, on behalf of the authors, I would like to warmly appreciate you again, and your specialist reviewers, for your time and consideration on the third round of critical, technical, and scientific evaluation of our manuscript.
I believe that our manuscript (manuscript number: ijms-1359758) was concisely reviewed by the most knowledgeable and resourceful reviewers in the revision team of your journal.
Below, you see all of the comments and our responses. It is worthy to mention that all of the new changes are done in the content of the manuscript in the bold light purple color in the font (without any highlights) in order to be distinct from first and second round of revision. Additionally, all of the revisions were made, but the page number or line numbering in some points were normally changed after deletion/insertions.
According to the reviewers' comments, their final decisions, and our responses, hope these explanations and responses are enough to meet the criteria for eligibility for possible (final) publication.
I am impatiently looking forward to your last decision on our manuscript.
Sincerely yours.
Dr. Abdolreza Esmaeilzadeh
Corresponding author
Comments and Suggestions for Authors
Reviewer 1
Dear authors, I’m sorry to say that I have again read until page 8 (out of 32 pages) and found multiple grammatical and scientific errors. The writing is still of poor quality and I personally can’t recommend publication of this article in its present form. The entire manuscript requires extensive editing to make the scientific contents clear and understandable. Thank you for your understanding.
Dear Editor!!! Thanks for the comment. We highly respect your concern. In the third round of revision, we re-evaluated all of the manuscript in case of accuracy of grammar. All of the English improvement and grammatical errors were corrected by a native English-speaking scientist (immunologist) throughout the manuscript. Additionally, we rechecked the manuscript to synchronize the content in order to make the language of our manuscript closer to the real science. All in all, we would like to make you rest assured that we re-evaluated the manuscript for the third time which you see in light purple in bold font (without any highlights).
Line 78-79: These types of studies contribute us in order to reach the most effective therapeutics…the sentence needs rephrasing…you can replace “contribute” with “help”
Thanks for the comment. Mentioned word was replaced with proposed word (light purple in bold font without any highlights) as you requested.
Line 84-87: Regardingly, molecular immunobiology may provide a clinical basis for further investigations of the therapeutic applications of cytokines [6, 7]…a subject name can’t provide anything…studies on that field can.
Thanks for the comment. Mentioned sentence was completely re-written (light purple in bold font without any highlights) as you requested.
Line 92-93: Interleukin-1 (IL-1) superfamily are essential players in the interactions of both innate and adaptive immune systems…again Interleukin-1 (IL-1) superfamily can’t be essential…members of the Interleukin-1 (IL-1) superfamily (e.g., different cytokines) can be..
Thanks for the comment. Mentioned sentence was completely re-written (light purple in bold font without any highlights) as you requested.
Line 102-107: Thus, the administration of novel IL-38 gene-based immunotherapeutics can be addressed to support this evidence understanding the complete mechanisms of IL-38 function and it’s effects on different diseases [19-22]…complicated and confusing sentence.
Thanks for the comment. Mentioned sentence was completely re-written (light purple in bold font without any highlights) as you requested.
Line 111-112: In this review, we firstly provide information on immunobiology (the structure, receptors, mechanism of signaling, and functional roles) of IL-38… the sentence needs rephrasing.
Thanks for the comment. Mentioned sentence was completely re-phrased (light purple in bold font without any highlights) as you requested.
Line 120-121: We believe that results of such studies can open optimistic insights into more collaborations between physicians and laboratory scientists… the sentence need rephrasing
Thanks for the comment. Mentioned sentence was completely re-phrased (light purple in bold font without any highlights) as you requested.
Line 124-127: Regardingly, we believe that results of such studies bi-functionally can highlight the importance of pre-clinical and basic medical studies in more accurate medical decisions, and optimal clinical outcomes for clinicians…I don’t understand the meaning of this
Thanks for the comment. Mentioned sentence was completely re-phrased (light purple in bold font without any highlights) as you requested.
Line 127-129: Also, clinical administration or therapeutic usage of seroimmunobiomarkers with reduced side effects can hopefully increase life- expectancy and qualify patients’ lifestyles…I can understand the usage, but what do you mean by administration of seroimmunobiomarkers? Also, what is “qualify patients’ lifestyles”
Thanks for the comment. Mentioned sentence was completely re-phrased (light purple in bold font without any highlights) as you requested.
Line 162: what do you mean by “were totally included”?
Thanks for the comment. Mentioned sentence was completely re-written (light purple in bold font without any highlights) as you requested.
Line 167: Studies that investigated secretion of IL-38 after usage of immunostimulants like vaccine, excluded…briefly mention why you excluded these study reports.
Thanks for the comment. Mentioned sentence was completely re-phrased (light purple in bold font without any highlights) as you requested. Because we wanted to report immunomodulation by only IL-38 and no any other immunostimulants or adjuvants, in order to omit any bias on the results.
Line 221-222: IL-38 from apoptotic cells is not properly recognized…discuss a bit more on this.
Thanks for the comment. Mentioned sentence was completely described according to the results of that article (light purple in bold font without any highlights) as you requested.
Line 253: IL-38 is present in all types of vertebrates with evolved immune responses [38]…”with evolved immune responses”. I’m not sure if this term is scientifically correct!
Thanks for the comment. As I made a quick Google search, there are a wide range of contents on evolution of immune responses. But, we replaced the word with “developed immune responses” (light purple in bold font without any highlights) as you requested.
Line 254-256: Despite numerous approaches for purification and expression of IL-38 in bacterial systems, Yuan et al successfully reached their hypothesis by using a prokaryotic expression vector (pET-44) and a one-step Ni2+-agarose affinity chromatography in a relatively large quantity of a C-terminus tagged IL-38 [27]…confusing and incorrect statement.
Thanks for the comment. Mentioned sentence was completely investigated and corrected (light purple in bold font without any highlights) as you requested.
Line 263: Additionally, IL-38 is numerated as an antagonism of for IL-36…what does it mean?
Thanks for the comment. Mentioned sentence was completely re-written (light purple in bold font without any highlights) as you requested.
Line 264-266: The anti-inflammatory role of IL-38 is postulated according to the sharing homology with IL-1Ra (an IL-1R1A), and with IL-36RA, making IL-38 as an antagonists with IL-1R1, and IL-36, respectively (20)…confusing sentence.
Thanks for the comment. Mentioned sentence was completely re-phrased and re-written (light purple in bold font without any highlights) as you requested.
Submission Date
16 August 2021
Date of this review
25 Sep 2021 09:51:19

Reviewer 2 Report
Congrats for your efforts!
Author Response
Comments and Suggestions for Authors
Reviewer 2
Congrats for your efforts!
Thanks for the comment and your precise evaluation. Definitely, we did our best to include and explain the main contribution of this article to the future studies after doing the corrections. We would like to warmly appreciate your positive feedback.
Submission Date
16 August 2021
Date of this review
13 Sep 2021 07:21:02

Round 4
Reviewer 1 Report
Though after the review(s) the manuscript looks much matured but still several inconsistencies are in existence. The article still feels fragmentary to some extent. The manuscript has a whole lot of information, it's just the presentation (writing) that is holding it backward. I observed several small mistakes (mentioned below) throughout the manuscript. Many texts in the figures are not legible and the font sizes need to be bigger. Below are my comments:
Figure 1: Some letters in this figure are unreadable, make them clear
Figure 2: Like fig 1, several letters are unreadable in figure 2, make them clear
Figure 3. Same thing with figure 3. I can’t read any text in figure 3. Also, the thick black border made the figure very clumsy.
Line 289-290:
Although IL-38 is bioactive enough to be a full-length molecule, it lacks of an integrated N-terminus to function as much as a processed cytokine [61]…I don’t understand this. Please explain.
Line 299-300:
Among these receptors, three receptors as IL-36R, IL-1R1, and Interleukin-1 Receptor Accessory Protein-Like 1 (IL-1RAPL1) have shown great potentials to interact with IL-38 [55]…grammatical error, correct the sentence.
Line 409-410:
(RA) is a very common autoimmune disease with long-term chronic inflammation of the synovium, causing swollen, and painful joints in the wrist and hands….Any particular reason for mentioning only wrist and hand? The pain can occur in the knees, ankles, elbows, hips, and shoulders as well.
Line 445-446:
It is worthy to note that, unlike IL-36/IL-36Ra that are firstly going to be increased, the elevation of articular IL-38 is postponed and accurately occurred at the later phases of CIA pathogenesis.
Rephrase the sentence
Line 480-482:
Totally, they reported that IL-38 can be promisingly served as a novel immunotherapeutic strategy, and a screening serobiomarker for patients with OA [127].
Did you mean to say immunotherapeutic target?
Line 684-685:
Many studies have been conducted on the mechanisms of the inflammatory pathways in the disease and significant results have been obtained [151].
Did you mean this: Many studies have been conducted to understand the mechanisms of the inflammatory pathways in this disease and significant results have been obtained [151].
Line 760-764:
Previously, it has been interrogated that when IL-38 is topically administrated for injured corneas in an alkali-induced corneal neovascularization mouse model, IL-38 is capable to diminish the inflammation-induced angiogenesis through attenuating secretion of IL1β, IL-6, IL-8, and TNF-α cytokines, and diminished angiogenesis-related activities including (proliferation, migration, and tube formation of human retinal endothelial cells) [157]…This sentence is too complicated. Make it simple/divide it into parts.
Line 906-909:
In another study, plasma levels and gene expression of IL-38 in PBMCs samples were significantly increased in ST-Elevated Myocardial Infarction (STEMI) patients in a time-dependent manner, compared with the control group (peaked at 24 h). In addition, plasma IL-38 levels were dramatically reduced in hyper-perfusion patients who were treated, compared with the control group…
Replace ‘with’ with ‘to’
Line 1140:
In a study conducted on a population of Iraqi men, the authors studied the relationship among several factors as Vitamin E, IL-17, IL-37 and IL-38, and Chronic Hepatitis B (CHB)…rephrase the sentence
Author Response
Dear Editor-in-Chief in the International Journal of Molecular Science
Jovial Greetings for the day.
Hope this email finds you well.
First and foremost, on behalf of the authors, I would like to warmly appreciate you again, and your specialist reviewers, for your time and consideration on the fourth round of critical, technical, and scientific evaluation of our manuscript.
I believe that our manuscript (manuscript number: ijms-1359758) was concisely reviewed by the most knowledgeable and resourceful reviewers in the revision team of your journal.
Below, you see all of the comments and our responses. It is worthy to mention that all of the new changes are done in the content of the manuscript in the bold black color in the font (with blue highlights) in order to be distinct from first, second, and third round of revision. Additionally, all of the revisions were made, but the page number or line numbering in some points were normally changed after deletion/insertions.
According to the reviewers' comments, their final decisions, and our responses, hope these explanations and responses are enough to meet the criteria for eligibility for possible (final) publication.
I am impatiently looking forward to your last decision on our manuscript.
Sincerely yours.
Dr. Abdolreza Esmaeilzadeh
Corresponding author
Comments and Suggestions for Authors
Though after the review(s) the manuscript looks much matured but still several inconsistencies are in existence. The article still feels fragmentary to some extent. The manuscript has a whole lot of information, it's just the presentation (writing) that is holding it backward. I observed several small mistakes (mentioned below) throughout the manuscript.
Dear Editor!!! Appreciation for the comment and fourth round of revision. We highly respect your concern. In the fourth round of revision, we re-evaluated all of the manuscript in case of accuracy of scientific content, accuracy of grammar, cohesion, and coherency. Also, all of the English improvement and grammatical errors were corrected by a English-speaking researcher (immunologist) throughout the manuscript. Additionally, we rechecked the manuscript to synchronize the content compatible with the most recently published data (especially the ones related to COVID-19) in order to make the language of our manuscript closer to the real science. All in all, we would like to make you rest assured that we re-evaluated the manuscript for the fourth time which you see in black in bold font (with blue highlights).
Many texts in the figures are not legible and the font sizes need to be bigger. Below are my comments: Figure 1: Some letters in this figure are unreadable, make them clear
Figure 2: Like fig 1, several letters are unreadable in figure 2, make them clear
Figure 3. Same thing with figure 3. I can’t read any text in figure 3. Also, the thick black border made the figure very clumsy.
Thanks for your comprehensive views. Those figures were re-designed with several different softwares and after changing fonts, size, contrast, colors, the pictures with the best-quality were chosen. We hope that they are of proper quality for publication.
Line 289-290:
Although IL-38 is bioactive enough to be a full-length molecule, it lacks of an integrated N-terminus to function as much as a processed cytokine [61]…I don’t understand this. Please explain.
Thanks for the comment. According to the biological data reported in that reference, mentioned sentence was completely rephrased and replaced with the best words describing the content (black in bold font with blue highlights) as you requested.
Line 299-300:
Among these receptors, three receptors as IL-36R, IL-1R1, and Interleukin-1 Receptor Accessory Protein-Like 1 (IL-1RAPL1) have shown great potentials to interact with IL-38 [55]…grammatical error, correct the sentence.
Thanks for the comment. Mentioned sentence was completely rephrased and replaced with the best words describing the content after checking grammatical errors with an English-speaking researcher (black in bold font with blue highlights) as you requested.
Line 409-410:
(RA) is a very common autoimmune disease with long-term chronic inflammation of the synovium, causing swollen, and painful joints in the wrist and hands….Any particular reason for mentioning only wrist and hand? The pain can occur in the knees, ankles, elbows, hips, and shoulders as well.
Many thanks for this comment. After a quick search, those aforementioned updates were added to this sentence, and totally, mentioned sentence was completely re-written (black in bold font with blue highlights) as you requested.
Line 445-446:
It is worthy to note that, unlike IL-36/IL-36Ra that are firstly going to be increased, the elevation of articular IL-38 is postponed and accurately occurred at the later phases of CIA pathogenesis. Rephrase the sentence.
Thanks for the comment. Mentioned sentence was completely rephrased and replaced with the best words describing the content (black in bold font with blue highlights) as you requested.
Line 480-482:
Totally, they reported that IL-38 can be promisingly served as a novel immunotherapeutic strategy, and a screening serobiomarker for patients with OA [127].
Did you mean to say immunotherapeutic target?
Thanks for the comment. Yes, exactly we meant it. Mentioned sentence was completely replaced with your proposed sentence (black in bold font with blue highlights) and even throughout the manuscript as you requested.
Line 684-685:
Many studies have been conducted on the mechanisms of the inflammatory pathways in the disease and significant results have been obtained [151].
Did you mean this: Many studies have been conducted to understand the mechanisms of the inflammatory pathways in this disease and significant results have been obtained [151].
Thanks for the comment. Yes, exactly we meant it. Mentioned sentence was completely replaced with your proposed sentence (black in bold font with blue highlights) as you requested.
Line 760-764:
Previously, it has been interrogated that when IL-38 is topically administrated for injured corneas in an alkali-induced corneal neovascularization mouse model, IL-38 is capable to diminish the inflammation-induced angiogenesis through attenuating secretion of IL1β, IL-6, IL-8, and TNF-α cytokines, and diminished angiogenesis-related activities including (proliferation, migration, and tube formation of human retinal endothelial cells) [157]…This sentence is too complicated. Make it simple/divide it into parts.
Thanks for the comment. Mentioned paragraph was completely rephrased, shortened, and replaced with the best and the simplest words describing the content in several sentences (black in bold font with blue highlights) as you requested.
Line 906-909:
In another study, plasma levels and gene expression of IL-38 in PBMCs samples were significantly increased in ST-Elevated Myocardial Infarction (STEMI) patients in a time-dependent manner, compared with the control group (peaked at 24 h). In addition, plasma IL-38 levels were dramatically reduced in hyper-perfusion patients who were treated, compared with the control group…
Replace ‘with’ with ‘to’.
Thanks for the comment. Mentioned word was replaced with proposed word (black in bold font with blue highlights) as you requested.
Line 1140:
In a study conducted on a population of Iraqi men, the authors studied the relationship among several factors as Vitamin E, IL-17, IL-37 and IL-38, and Chronic Hepatitis B (CHB)…rephrase the sentence.
Thanks for the comment. Mentioned paragraph was completely rephrased and replaced with the best words describing the content (black in bold font with blue highlights) as you requested.
Submission Date
16 August 2021
Date of this review
04 Oct 2021 08:42:37
